# Phenotypes on demand via switchable target protein degradation in multicellular organisms

Frederik Faden[1,2], Thomas Ramezani[3,4,†], Stefan Mielke[1,2], Isabel Almudi[5,†], Knud Nairz[5,†], Marceli S. Froehlich[6,†], Jörg Höckendorff[6,†], Wolfgang Brandt[7], Wolfgang Hoehenwarter[8], R. Jürgen Dohmen[6], Arp Schnittger[3,4,9,†] & Nico Dissmeyer[1,2,3,4,9]

Phenotypes on-demand generated by controlling activation and accumulation of proteins of interest are invaluable tools to analyse and engineer biological processes. While temperature-sensitive alleles are frequently used as conditional mutants in microorganisms, they are usually difficult to identify in multicellular species. Here we present a versatile and transferable, genetically stable system based on a low-temperature-controlled N-terminal degradation signal (lt-degron) that allows reversible and switch-like tuning of protein levels under physiological conditions *in vivo*. Thereby, developmental effects can be triggered and phenotypes on demand generated. The lt-degron was established to produce conditional and cell-type-specific phenotypes and is generally applicable in a wide range of organisms, from eukaryotic microorganisms to plants and poikilothermic animals. We have successfully applied this system to control the abundance and function of transcription factors and different enzymes by tunable protein accumulation.

[1] Independent Junior Research Group on Protein Recognition and Degradation, Leibniz Institute of Plant Biochemistry (IPB), Weinberg 3, D-06120 Halle (Saale), Germany. [2] ScienceCampus Halle—Plant-based Bioeconomy, Betty-Heimann-Strasse 3, D-06120 Halle (Saale), Germany. [3] University Group at the Max Planck Institute for Plant Breeding Research (MPIPZ), Max Delbrück Laboratory, Carl-von-Linné-Weg 10, D-50829 Cologne, Germany. [4] University of Cologne, Institute of Botany III, Biocenter , Zülpicher Str. 47 b, D-50674 Cologne, Germany. [5] Institute of Molecular Systems Biology (IMSB), Swiss Federal Institute of Technology (ETH), Wolfgang-Pauli-Strasse 16, CH-8093 Zurich, Switzerland. [6] Institute for Genetics, Biocenter, University of Cologne, Zülpicher Straße 47a, D-50674 Cologne, Germany. [7] Computational Chemistry, Department of Bioorganic Chemistry, Leibniz Institute of Plant Biochemistry (IPB), Weinberg 3, D-06120 Halle (Saale), Germany. [8] Proteomics Unit, Leibniz Institute of Plant Biochemistry (IPB), Weinberg 3, Halle (Saale) D-06120, Germany. [9] Département Mécanismes Moléculaires de la Plasticité Phénotypique, Institut de Biologie Moléculaire des Plantes du CNRS, IBMP-CNRS, Unité Propre de Recherche 2357, Conventionné avec l'Université de Strasbourg, 12, rue du Général Zimmer, Strasbourg F-67000, France. † Present addresses: The Queen's Medical Research Institute, MRC Centre for Inflammation Research, The University of Edinburgh, 47 Little France Crescent, Edinburgh EH16 4TJ, UK (T.R.); Andalusian Centre for Developmental Biology (CABD), Universidad Pablo de Olavide, Ctra. Utrera Km. 1, 41013 Seville, Spain (I.A.); Novino Slowdrink GmbH, Blümlisalpstrasse 31, CH-8006 Zürich, Switzerland and Inselspital Bern, Freiburgstrasse 10, CH-3010 Bern, Switzerland (K.N.); Grünenthal Innovation—Early Clinical Development, Grünenthal GmbH, Zieglerstraße 6, D-52078 Aachen, Germany (M.S.F.); UCB BIOSCIENCES GmbH, Alfred-Nobel-Straße 10, D-40789 Monheim, Germany (J.H.); Department of Developmental Biology, Biozentrum Klein Flottbek, University of Hamburg, Ohnhorststrasse 18, D-22609 Hamburg, Germany (A.S.). Correspondence and requests for materials should be addressed to N.D. (email: nico.dissmeyer@ipb-halle.de).

The classical way to generate temperature-sensitive alleles comprises random mutagenesis followed by large-scale screens at different temperatures. This procedure is usually limited to fast-growing single-celled organisms where mutant populations can be analysed simultaneously at restrictive and permissive temperatures. Major problems in using temperature-sensitive alleles are their generation, identification, establishment and leakiness. An alternative way to generate heat-sensitive mutants is to use a temperature-inducible N-degron (td), where a protein of interest (POI) is fused to a portable N-terminal degradation cassette[1]. However, to date this technique has been solely used in unicellular eukaryotes in a temperature range impractical for most multicellular organisms. We show how to efficiently circumvent these limitations by using a novel portable td fusion protein to render the levels of active POIs conditional.

Generally, degradation signals within a protein sequence that make it short-lived in vivo or in vitro are called 'degrons'[2], and an 'N-degron'[3] is an amino-terminal (N-terminal) degron, which relates the metabolic stability of a protein to the identity of its amino-terminal residue depending on the N-end rule pathway[4]. The N-end rule pathway is part of the ubiquitin (Ub)/proteasome system and has been shown to be active in yeast, animals and plants[5–9]. It maintains proteostasis as a protein quality control mechanism by removing cleaved, damaged or misfolded proteins from the cell[7,10,11]. N-degrons comprise several determinants to target a substrate to the N-end rule pathway. First, they must contain a destabilizing N-terminal amino acid that can be recognized by N-end rule pathway-specific E3 Ub ligases (N-recognins). Second, another crucial factor is a certain flexibility and accessibility of the N-terminal amino acid enabling a proper recognition of the substrate[12]. Third, substrates need to contain at least one internal Lys in appropriate distance to the N terminus, which may serve as polyubiquitination site[3]. The N-end rule pathway targets both cytosolic and nuclear substrates but also proteins localized in the membrane[13].

Conditional protein expression via heat-sensitive N-degrons was designed as a genetic tool to generate conditional, temperature-inducible alleles in budding yeast[1]. The system is based on rapid, reversible depletion or accumulation of POIs and targets the entire protein for proteasome-dependent proteolysis at a restrictive temperature. Therefore, it can be used as a rapid ON/OFF system for reversible accumulation of active proteins (Fig. 1a,b and Supplementary Fig. 1). Here an N-degron serves as a destabilizing N-terminal tag (26 kDa) fused to the POI as a portable td cassette. It consists of three vital parts: (1) Ub that is co-translationally removed from the fusion by deubiquitinating enzymes (Ub-fusion technique[14]); (2) a thermo-labile mouse dihydrofolate reductase ($DHFR^{ts}$) that triggers protein degradation at the restrictive (high) temperature; and (3) a destabilizing N-terminal residue at the $Ub–DHFR^{ts}$ junction, which is exposed after deubiquitination the full fusion construct, which serves as dormant N-degron and can be recognized by N-end rule E3 Ub ligases (Supplementary Fig. 1).

Even if a potential degron (that is, a motif theoretically recognized by a N-end rule pathway E3 Ub ligase) is present on the surface of a target, its conformational rigidity may prevent its recognition due to steric hindrance. A conformationally destabilized protein may contain formerly buried (cryptic) surface degrons (that is, dormant degrons), which were masked in a stabilized version of the protein[15]. In the classical yeast td degron, both happens according to the presented model[1]: a previously rigid surface degron (N-terminal) undergoes conformational relaxation and previously buried degrons, such as internal Lys side chains, are exposed.

However, to date, this system has only been used at the single-cell level in yeast and cell culture where it requires restrictive temperatures of 37 or 42 °C (refs 1,16,17). These high temperatures are beyond the physiological range of many multicellular organisms, including animals and plants[18]. Although there are alternative methods for conditional protein shut-off via degradation, for example, the protease-mediated TIPI[19], the destabilizing domain (DD) systems[20] and the auxin-inducible degron (AID)[21], their use is also limited to cells in culture[22]. The latter two systems require the addition of exogenous compounds to trigger the response (Supplementary Note 1).

To fill the gap of a conditional protein accumulation system for multicellular systems under physiological conditions, we constructed a low-temperature (lt) N-degron and show its general applicability by studying seven POIs in five different model systems (that is, two plant species (Arabidopsis thaliana and Nicotiana benthamiana (tobacco)), Drosophila melanogaster (fruit fly), Drosophila cell culture and Saccharomyces cerevisiae (budding yeast). The design of the lt-degron not only allowed in vivo inactivation and depletion of proteins and enzymes but also tuning of protein levels and enzyme activity. Depending on the POI, we demonstrated triggering of developmental processes on demand. We define the temporal requirement of a transcriptional regulator protein mandatory for the formation of trichomes, which belong to the prime model cells in plant developmental genetics. Moreover, we made use of a developmental switch by stabilizing a transcription factor causing flowering on demand, triggered protein accumulation in N. benthamiana (tobacco), conditionally depleted proteins in cell culture and Drosophila flies. Therefore, the link to in-depth applications in developmental studies is addressed as well as the proof of concept for applications of the lt-degron in biotechnologically relevant contexts.

## Results

**N-degron functionality in multicellular organisms.** To assess N-degron functionality in multicellular organisms, we chose easily scorable, quantitative and irreversible biological read-outs after temperature shifts in vivo, like the development of Arabidopsis trichomes (leaf hairs) and flower induction. We first asked whether the established yeast td degron containing the single-mutated $DHFR^{P67L}$ (termed 'K1'), exhibiting a restrictive temperature of 37–42 °C (refs 1,16,17,23) is functional in plants. Our DHFR was initiated by phenylalanine (Phe), a strongly destabilizing residue in plants, which is presumably recognized by the bona fide N-end rule E3 Ub ligase PROTEOLYSIS1 (PRT1)[24–26]. As the first test system, we used the formation of Arabidopsis trichomes, which are a well-established model in cell and developmental biology[27–29], and an attractive target for plant metabolic engineering.[30] WD40 protein TRANSPARENT TESTA GLABRA1 (TTG1) is an essential regulator for trichome development[31], ttg1 mutants are devoid of trichomes (Figs 1c and 2a) but the mutant phenotype can be rescued by expressing TTG1 under control of the constitutive cauliflower mosaic virus (CaMV) 35S promoter (Pro35S)[32]. In this evaluation system, the read-out for a functional N-degron is the number of trichomes formed per leaf after the shift from permissive to restrictive temperature. Under permissive conditions, the degron is non-functional, that is, the POI will accumulate and form trichomes, whereas under restrictive conditions, the degron is active and leads to the degradation of the fusion protein, and leaves will be devoid of trichomes.

To this end, transgenic ttg1 mutant Arabidopsis plants were generated that expressed TTG1 fused to degron cassette K1 (Supplementary Fig. 2). We set the presumptive maximal restrictive temperature treatment to 29 °C, which is near but not above the upper end of the physiological temperature range of

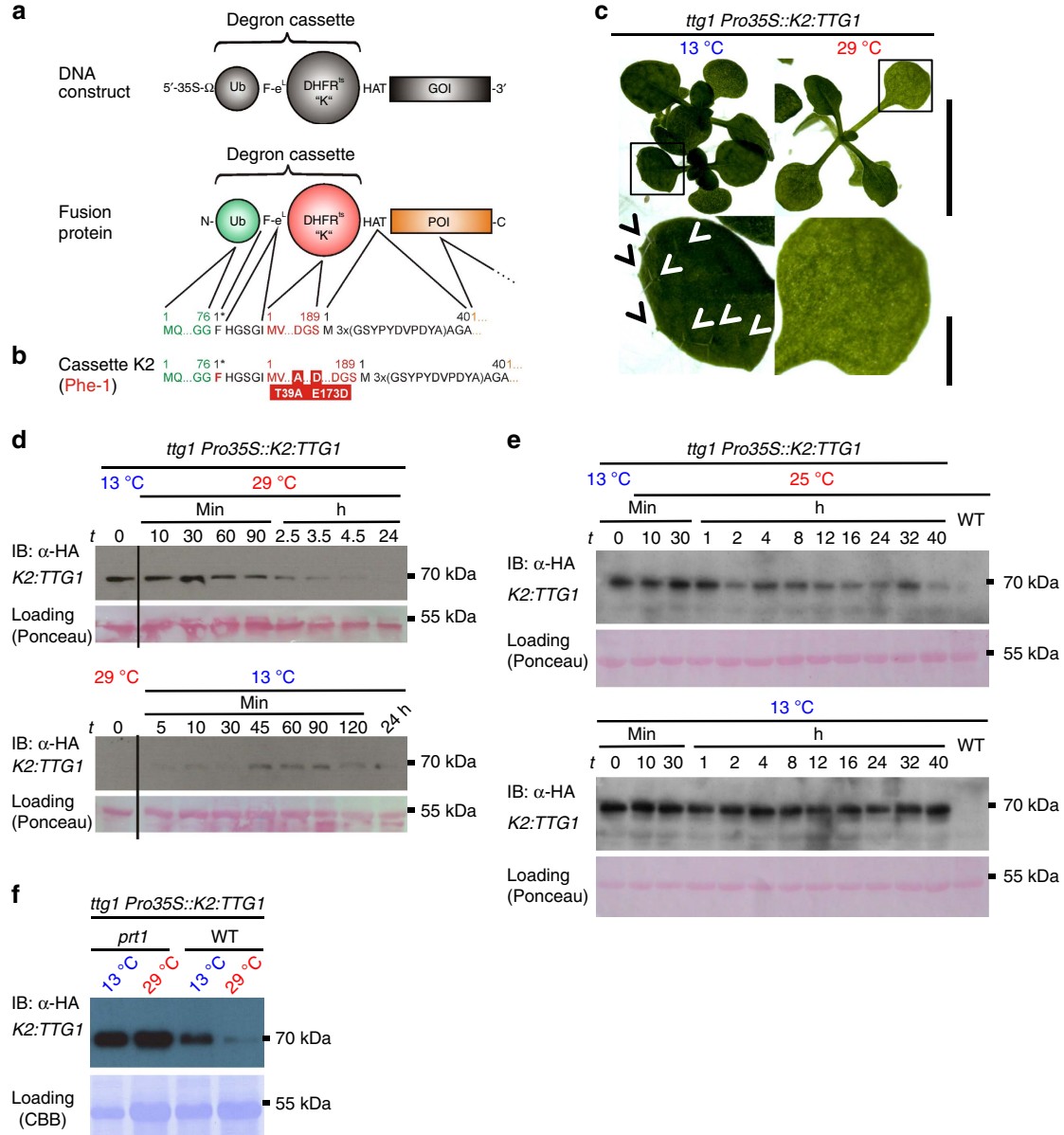

**Figure 1 | Conditional protein depletion/accumulation cause developmental phenotypes on demand *in vivo* and can be used to develop trichome cells.** (**a,b**) Lt-degron architecture. The mechanism of conditional N-degron containing protein depletion is shown in Supplementary Fig. 1, degron construct design is shown in Supplementary Fig. 2. (**c**) *ttg1 K2:TTG1* plants grown at permissive or restrictive temperatures. Arrow heads indicate trichomes in inlets. Scale bar, 1 cm, 5 mm (inlet). (**d**) *In vivo* protein depletion time course of *K2:TTG1* from transgenic *ttg1 Pro35S::K2:TTG1* mutant plants. (**e**) *In vivo* dimming of *K2:TTG1* levels by application of semi-permissive temperatures. Plants were subjected to temperature-shift experiments from 13 to 25 °C. Western blots of plants shifted to semi-restrictive (25 °C) and kept at constitutively permissive temperature (13 °C) are shown. Plants were shifted at $t = 0$ from 13 to 25 °C (upper panel) or kept at 13 °C (lower panel). (**f**) Stabilization of *K2:TTG* in *prt1*. The *K2:TTG* fusion protein stabilizes in the *prt1* mutant background under restrictive temperature indicating that the E3 ligase PRT1 is the principal component responsible for *K2* degradation. *K2:TTG* also partially accumulates under permissive temperatures in the *prt1* mutant background compared with the WT indicating that also here, the fusion protein is not fully stabilized. All samples derived from the same experiment, processed in parallel and leaves of the same developmental stage were compared. Data were confirmed by analysis of at least three biological replicates. Equal loading was further confirmed by Ponceau S staining of blotted and immunostained membranes.

*Arabidopsis*[18] and found that *K1:TTG1* stably transformed plants developed trichomes also at this temperature indicating that the fusion protein is not sufficiently destabilized (Supplementary Table 1).

To lower the restrictive temperature, we generated an N-degron variant containing a triple mutated *DHFR*[T39A/P67L/E173D] (*K3*) with two additional substitutions (T39A/E173D). We had previously identified the latter mutations in a yeast mutagenesis

screen for destabilizing point mutations within *DHFR* (Supplementary Note 2)[33]. However, *ttg1* mutants containing *K3:TTG1* never formed trichomes under any tested conditions. The protein appeared to be unstable after growing the plants at constitutively permissive or restrictive temperatures or shifting them between the two conditions (Supplementary Fig. 3 and Supplementary Table 1). Lack of trichomes under low temperatures correlated with levels of *K3:TTG1* protein undetectable by western blot

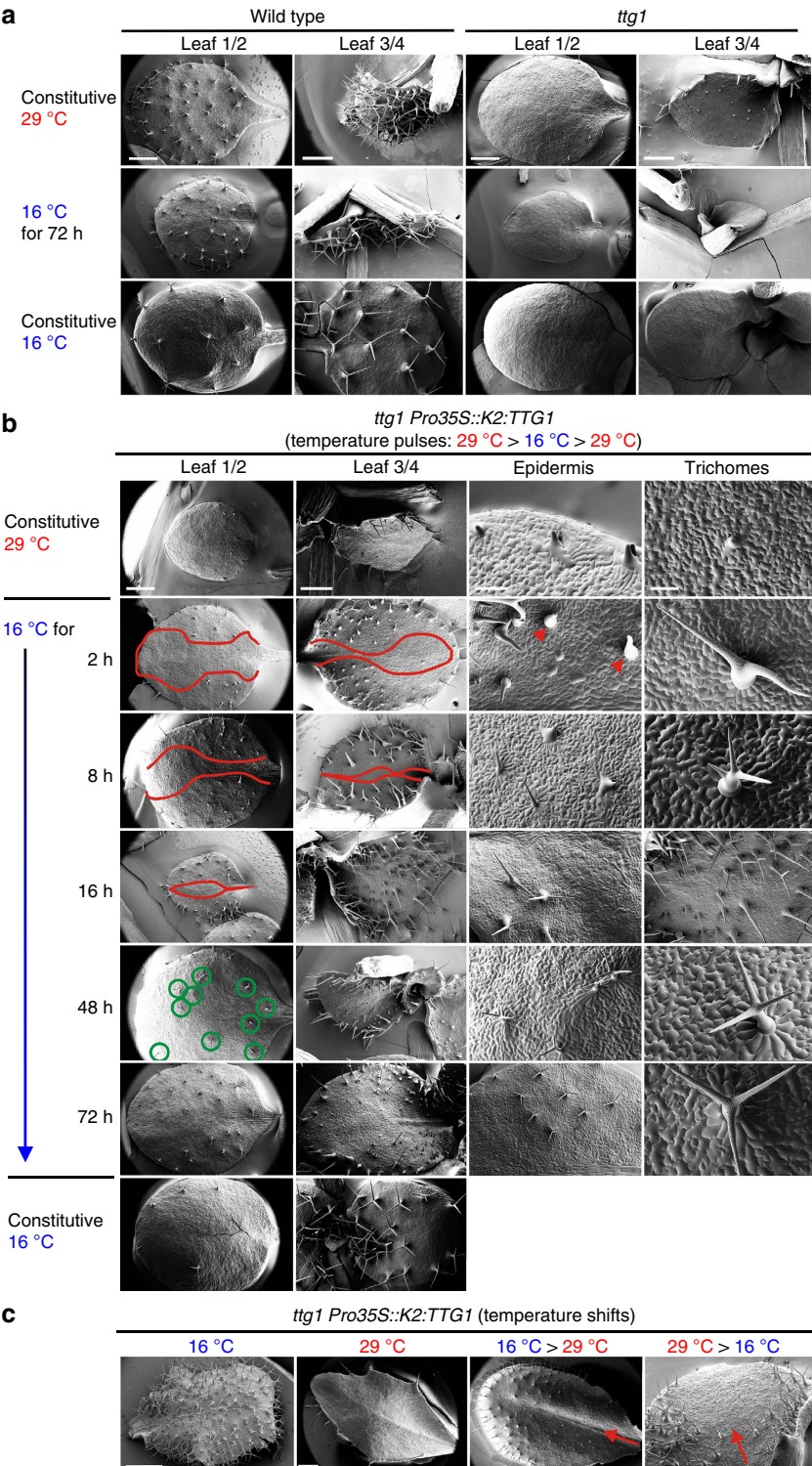

**Figure 2 | Spatiotemporal development of trichomes as test system for conditional protein accumulation.** (**a**) Phenotype of *ttg1* mutants compared with the WT. Cryo scanning electron micrographs were taken from 4-week-old plants grown constitutively at 29 or 16 °C. Scale bar, 1 mm. (**b,c**) Conditional complementation of *ttg1* with *K2:TTG1* depending on temperature and time. SEMs of 4-week-old *ttg1 K2:TTG1* plants. (**b**) Temperature-pulsing of *K2:TTG1* protein in *ttg1* by short shifts from the restrictive to the permissive temperature (16 °C). Control plants were grown constitutively at restrictive (29 °C, upper row) or permissive (16 °C, bottom row) temperatures. Centre: plants grown at restrictive and shifted to permissive temperature for the time indicated (2–48 h). Red lines: borders of trichome initiation zone moving from lateral to medial with duration of the *K2:TTG1* pulse. Green circles: fully mature trichomes and epidermal patterning appearing after 48 h. Arrow heads: loped circumference at trichome bases. Scale bars, 1 mm (left), 500 µm (second and third columns) and 100 µm (right). (**c**) WT- and *ttg1*-like glabrous leaves of *ttg1 K2:TTG1* grown at permissive and restrictive temperatures (left two panels) or after temperature up- and downshifts during leaf development (right two panels). Arrows: direction of proximo-distal leaf development highlighting the sector affected by the temperature shift. A phenotypic comparison of different degron variants is shown in Supplementary Fig. 3, expression levels of all degron fusion genes under restrictive and permissive conditions are presented in Supplementary Fig. 4. Molecular dynamics simulations and models of the DHFR variants used are presented in Supplementary Fig. 5.

analysis although its transcript was found (Supplementary Fig. 3). These findings indicated that proteins carrying the *K3* degron are too unstable at lower temperatures to provide a permissive phenotype.

**Novel lt N-degron for *in vivo* application.** Finally, we fused *TTG1* to *DHFR^T39A/E173D* (*K2*), which only contains the two destabilizing point mutations *T39A/E173D* identified in the mutagenesis screen. The expression of *K2:TTG1* caused a conditional (that is, temperature-dependent) restoration of trichomes in the *ttg1* mutant background (Figs 1c and 2b,c, Supplementary Fig. 3 and Supplementary Table 1). In time-course experiments, decreasing and increasing levels of *K2:TTG1* protein could be tracked as a consequence of temperature up- or downshifts causing depletion versus accumulation (Fig. 1d). Neither the endogenous TTG1 protein without the lt-degron nor the *K2:TTG1* transcript levels responded to the temperature shifts when tested in a transgenic *ttg1 TTG1-HA* complementation line (Supplementary Figs 3a and 4a).

For the classical temperature-induced yeast N-degron, it was hypothesized that after heat-induction, a dormant N-terminal degron becomes active after conformational changes and that internal Lys side chains get exposed[1]. We tested this hypothesis for the point mutations present in the sequences of the DHFR variants of the classical td (*K1*) and our novel lt-degron (*K2*) by stability predictions and molecular dynamics simulations (Supplementary Table 2 and Supplementary Note 3). Modelling of the point mutations within the DHFR crystal structure suggested that *E173D* contributes to a higher conformational flexibility and increases the exposure of Lys side chains on the surface under elevated temperatures. This may lead to a better accessibility to the ubiquitination machinery (Supplementary Fig. 5a–d).

**N-terminal accessibility and N-end rule requirements.** We hypothesized that a possible hyper-destabilization of *K3:TTG1* could be due to the flexibility introduced by the short linker e^L (2-His-Gly-Ser-Gly-Ile-6; Supplementary Figs 2,3) between DHFR and the destabilizing Phe residue compared with the classical yeast degron[34]. By deleting e^L from the degron cassettes and thus making the destabilizing N-terminal Phe less accessible for the N-end rule Ub ligases, we tested if a conditional phenotype can now be observed as a way to further optimize, that is, decrease, the restrictive temperature of the system. However, transgenic plants expressing *K3:TTG1* lacking e^L showed very similar phenotypes as the previously tested e^L-containing versions. Similarly, only expression of *K2:CO* containing the linker led to a temperature-dependent response, the constructs lacking the linker did not (Supplementary Table 1 and Supplementary Fig. 3).

As control, cassette *K4* was engineered. It contains the same DHFR variant as *K2* but starts with a methionine and can therefore not be recognized by N-end rule E3 ligases due to the lacking destabilizing N-terminal residue (Supplementary Figs 2 and 3). The *T39A* and *E173D* mutations were previously shown to cause degradation by a protein quality control pathway at higher temperatures (30 °C)[33]. The *K4:TTG1* phenotypes were highly variable but not temperature-dependent as expected from a conditionally stable *TTG1-td* (Supplementary Fig. 3c,d and Supplementary Table 1). *K4:TTG1* plants developed trichomes regardless of the growth temperature. These data demonstrated that the destabilizing N-terminal Phe residue is a critical feature of the conditional lt *K2* degron.

**Tuning of protein levels.** Indeed, protein accumulation by this novel lt-degron containing the *K2* cassette was even tuneable

and allowed fading out the POI to achieve intermediate levels at semi-restrictive temperatures (Fig. 1e). For this experiment, plants were shifted at $t = 0$ from 13 to 25 °C or kept at 13 °C. Protein degradation appeared slower when shifted to 25 °C compared with restrictive temperature (Fig. 1d), and low levels of POI were still detectable after 40 h. The decrease of lt-protein over time could be observed reaching a level lower than at permissive but higher than at restrictive temperatures. To test for the requirement of a functional N-end rule pathway in the host organism to efficiently degrade the Phe-initiated lt-degron, we introgressed *ttg1 K2:TTG* into the *prt1* mutant. Proteins with initiating Phe at their N terminus are expected to be recognized by one of the two known plant N-recognins, namely PRT1. We observed that the fusion protein is stabile in the *prt1* mutant background also at restrictive temperature (Fig. 1f).

**Triggering trichome induction on demand.** The tunability of *K2:TTG1* and the conditional induction of trichomes in the mutant background allowed us to perform previously impossible experiments, that is, to dissect the temporal requirement of TTG1 function during trichome establishment and maturation in great detail (Fig. 2b,c). Taking the pattern of wild-type (WT) trichome distribution into account[35], we found that a pulse of *K2:TTG1* protein is needed for at least 2 h to establish a trichome fate during the initial induction phase (Fig. 2b,c). After a pulse of 2 h at permissive temperature of 16 °C, the front of the trichome initiation sites started to move proximal to the midvein and trichomes began to differentiate (branching). Notably, initiation sites develop proximal to the mid-vein as spots with enlarged precursor cells, suggesting that different leaf areas have a differential predisposition to form trichomes. After 8 h, further branch points are formed and few three-branched, albeit not fully developed, trichomes appeared. Many trichomes formed a loped circumference, which is typical for de-differentiation if initial maintenance of cell fate is not completed[36]. After 16 h, a spacing pattern was formed indicating robust trichome maintenance. Forty-eight hours of TTG1 function were needed for developing WT-like three-branched trichomes. Expansion and partial restoration of a WT-like distribution pattern, as well as morphology of trichomes, is fully accomplished after 72 h of TTG1 action (Fig. 2b). Here the lt-degron allowed us to determine the temporal requirement of TTG1 in the context of the development of a highly specific cell type.

**Switching floral induction on demand.** As a second test system with an easily scorable biological read-out, we chose WT *Arabidopsis* plants that conditionally overexpress the transcription factor CONSTANS (CO), a key regulator of flowering time[37]. Overexpressing *CO* causes a rapid shift from the vegetative to reproductive growth mode and thereby an early flowering[38,39]. Thus, at permissive temperatures, plants expressing a functional degron–CO fusion protein (*CO-td*) were expected to flower earlier than plants grown under the restrictive temperature. The read-out is hence the days till flowering at permissive temperature in comparison with the WT. Similarly to the *TTG1-td* constructs, neither *K1:CO* nor *K3:CO* showed a temperature-dependent phenotype (Supplementary Table 1). In contrast, the lt-degron system very efficiently worked as expression of *K2:CO* at the permissive temperature caused early floral induction. Here the inducible formation of flower meristems occurred about 14 days earlier than in the WT when grown at the same permissive temperature (Fig. 3a). No differences were found when plants were grown at the restrictive temperature as in a time-course experiment, the onset of flowering was observed from day 15 onwards for both WT and *K2:CO* plants (Supplementary Fig. 6).

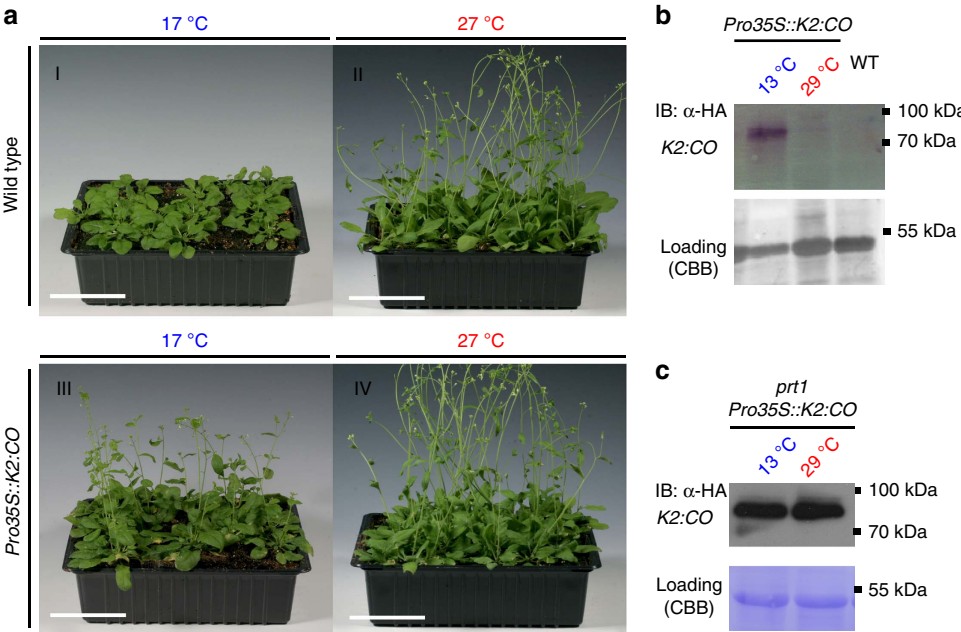

**Figure 3 | Switchable flower meristems by conditional accumulation of *CO-td*.** (**a**) Conditional expression of *CO-td* fusion leads to premature flowering *in vivo*. Four-week-old (I and II) WT and (III and IV) *K2:CO* transgenic plants grown under permissive and restrictive temperatures and short-day conditions. (III) In transgenic *K2:CO* plants, plants bolted earlier than in the WT (I). Scale bars, 5 cm. (**b**) *K2:CO* protein levels from transgenic plants grown at permissive or restrictive temperature. (**c**) Stabilization of *K2:CO* in *prt1*. The *K2:CO* fusion protein stabilizes in the *prt1* mutant background under restrictive temperature indicating that the E3 ligase PRT1 is required for *K2* degradation. For details on N-degron technology, see Supplementary Fig. 1; for construct layout, see Supplementary Fig. 2. For transcript levels under restrictive and permissive conditions, see Supplementary Fig. 4. A developmental time-course experiment for *CO-td* is shown in Supplementary Fig. 6.

This phenotype correlates with decreased levels of *K2:CO* under restrictive conditions while transcript levels remained unchanged (Fig. 3b and Supplementary Fig. 4b). *K2:CO* introgressed into *prt1* lead to stabilization of the fusion protein also at restrictive temperature (Fig. 3c). Here conditionally stable *CO-td*, enabled us to control the onset of flowering and to generate floral meristem cells, that is, specialized cell types, on demand.

***In vivo* inactivation and depletion of proteins and enzymes**. To further assess the application spectrum of the lt-degron in plants, we chose green fluorescent protein (GFP), β-glucuronidase (GUS) and tobacco etch virus protease (TEV) as additional target proteins to be tested when stably transformed in *Arabidopsis*. *K2:GFP* robustly followed the accumulation/depletion regime (Fig. 4a,b and Supplementary Fig. 4c). Moreover, for *K2:GFP*, we have designed two different lt-degron cassettes to test for degradation via two known branches of the N-end rule pathway. We used either the previously mentioned Phe or, as another potentially destabilizing residue, Arg, which results as neo-N-terminal after deubiquitination of the translated *K2:GFP* fusion protein. These residues are expected to be recognized by the two known N-recognins, that is, Arg via PRT6 and Phe via PRT1. The response of these two constructs was compared under standard growth conditions (Supplementary Fig. 4c) and in time-course experiments (Supplementary Fig. 4d) to demonstrate robustness. Both destabilizing N termini gave comparable, strongly temperature-dependent results and revealed that the lt-degron is functional by both branches of the N-end rule pathway.

Histological staining for *K2:GUS* revealed activity of the fusion protein *in vivo* at permissive and a significantly reduced activity at restrictive temperature (Fig. 4c). *K2:GUS* protein levels and activity were strongly decreased for individuals grown constitutively at restrictive temperature (Fig. 4d,e) with transcript levels unchanged (Supplementary Fig. 4e). In time-course experiments,

*K2:GUS* protein levels increased and correlated with high enzyme activity after shift to permissive temperature and vice versa if shifted to restrictive conditions (Fig. 4f,g). *K2:GUS* western blots showed two bands of different sizes (Fig. 4d,f). To identify the *K2:GUS* population with a temperature-dependent hydrolase activity, which accumulates at permissive temperature, we analysed the bands by mass spectrometry (Supplementary Fig. 7). This revealed that only the *K2:GUS* population, which accumulated at the permissive temperature contains the lt-degron part and is highly active (Fig. 4e,g). At the same time, *K2:GUS* is stabilized also at restrictive temperature after the addition of proteasome inhibitor indicating the degradation of the fusion protein via the Ub/proteasome system (Supplementary Fig. 4f).

**Tuning of activity levels**. To further strengthen our claim on the temperature-dependent tunability of the lt-degron system, we performed temperature-shift assays with *ProUBQ10::K2:GUS* addressing temperature-dependent changes in both protein and activity levels at various temperatures (Fig. 4h). In contrast to the *in vivo* dimming of *K2:TTG1* (Fig. 1e), this represents a more appropriate tuning experiment because here, the activity of the conditional lt-POI could be quantified in relation to the lt-POI protein levels and total protein content of the sample. The levels of active *K2:GUS* fusion protein were stabilized and its activity, as monitored by the quantitative GUS assay, followed the increased or decreased protein levels according to the applied temperatures in the permissive to restrictive range. Moreover, *K2:GUS* activity levels can be efficiently regulated by means of different temperatures. All of this indicated a high flexibility of the lt-degron regardless of temperature adjustment from permissive to restrictive temperature or vice versa.

**Application of lt-degrons in *N. benthamiana***. Next, we applied our low-temperature approach to TEV as a target, which is widely

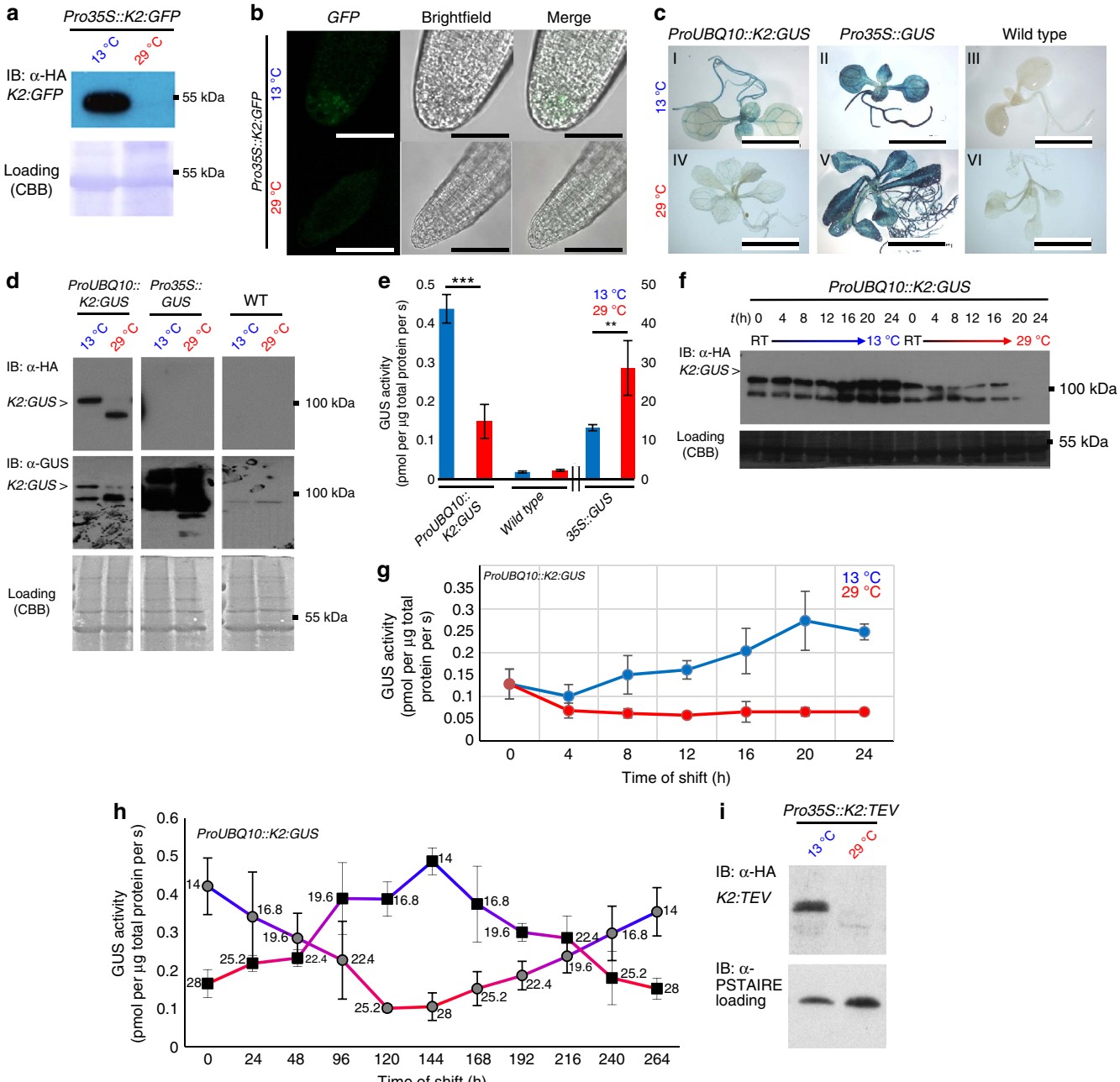

**Figure 4 | Conditional molecular phenotypes are linked with protein degradation and deactivation of lt-degron fusion proteins. (a)** Conditional accumulation of *K2:GFP* fusion protein in whole seedlings and (**b**) functionality of *K2:GFP* in root tips. Also Arg-initiated *K2:GFP* is conditionally depleted (Supplementary Fig. 4c). Scale bar, 50 μm. (**c**) *In vivo* hydrolase activity of transgenic *ProUBQ10::K2:GUS* compared with *Pro35S::GUS*, overexpression lines and the WT constitutively grown at permissive or restrictive temperature. Scale bar, 1 cm. (**d**) *K2:GUS* protein levels of seedlings constitutively grown at the indicated temperatures. Membranes were immunostained with α-HA antibody followed by a second staining with α-GUS antibody after stripping. The asterisk indicates *K2:GUS* protein. (**e**) Quantitative GUS assays of plants constitutively grown at 13 versus 29 °C. Six biological replicates were performed for *K2:GUS* and three biological replicates for WT and *35S::GUS*. All samples were measured as three technical replicates. *P* values for Student's *t*-test (two-sided *t*-test, type 3) are *K2:GUS*: $6.44766 \times 10^{-7}$ (***); *35S::GUS*: 0.004365663 (**). All error bars in this panel represent s.d., asterisk: significant. (**f**) Western blots of *ProUBQ10::K2:GUS* seedlings constitutively grown at ambient temperature (room temperature (RT)) shifted to permissive or restrictive temperature. (**g**) Quantitation of enzyme activity of **f**. Seedlings were grown under standard long-day conditions (16-h light, 8-h dark, 21 °C) aseptically on MS plates. One-week-old seedlings were shifted to permissive or restrictive conditions, respectively. In all, 5–10 individual seedlings were collected and pooled every 4 h. Three biological replicates were performed with two technical replicates each. All error bars in this panel represent s.d. (**h**) Degron tuning assay monitoring *K2:GUS* activity through temperature shift. Plants expressing *ProUBQ10::K2:GUS* were grown at cold (permissive) or warm (restrictive) temperature and then shifted to the respective opposite temperature and back over the course of 10 days. Samples were taken every 24 h, followed by a temperature shift of 2.8 °C. Activity of the *K2:GUS* fusion protein clearly follows the temperature shift. The numbers next to the data points represent the sampling temperature, *n* = 3. All error bars in this panel represent s.d. MS analysis of *K2:GUS* populations is documented in Supplementary Fig. 7. (**i**) *K2:TEV* in stably transformed *Arabidopsis* plants. Western blot of material collected from seedlings grown at permissive and restrictive temperatures. Equal loading was further confirmed by staining of blotted membranes with Coomassie Brilliant Blue G225 after immunostaining or α-PSTAIRE antibody detecting CYCLIN-DEPENDENT KINASE A;1 (CDKA;1). For transcript levels under restrictive and permissive conditions, see Supplementary Fig 4.

used for cleaving proteins *in vitro* and *in vivo*. The protease can be expressed for site-specific protein cleavage in many different host organisms without causing adverse effects[40]. In *Arabidopsis*, *K2:TEV* transcript levels were unaffected by temperature shifts (Supplementary Fig. 4g), but the fusion protein accumulated exclusively at permissive temperature (Fig. 4i).

*N. benthamiana* (tobacco) plants are widely used expression hosts for plant-made pharmaceuticals and high-value proteins. Therefore, we tested the lt-degron in tobacco, which was transiently transformed with *K2:TEV*. The fusion protein was depleted from plants shifted to 29 °C and accumulated in individuals grown at 13 °C (Fig. 5a). A conditional TEV offers opportunities for downstream processing of recombinant proteins by intracellular cleavage. In addition to *K2:TEV*, *K2:GUS* and a fusion of *K2* with the agriculturally important herbicide resistance proteins phosphinothricin-*N*-acetyltransferase (PAT) were also expressed in a conditional manner in tobacco (Fig. 5b,c). Together these findings established that the lt-degron approach was also suitable for the manipulation of tobacco and therefore is likely applicable for many more plant species.

### Conditional protein depletion in *S. cerevisiae*.

To test whether the lt-degron system is also functional outside of plant species, we used orotidine 5′-phosphate decarboxylase (URA3) as a reporter in *S. cerevisiae*. URA3 functionality and half-life were tested at 17.5 and 27 °C in the WT, in *ubr1-Δ* lacking the functional N-end rule pathway E3 Ub ligase UBR1 and in *ump1-Δ*, with impaired proteasome function (Fig. 6a). Experiments in WT (*JD47-13C*), *ubr1-Δ* (*JD55*) and *ump1-Δ* (*JD59*) strains revealed that *K1* and *K2* confer uracil prototrophic growth in the *ubr1-Δ* and *ump1-Δ* mutants at all temperatures that were generally growth-permissive for these strains, and at lower temperatures in WT cells (Supplementary Fig. 2c). Increasing the temperature, however, resulted in growth inhibition of the WT strain expressing these constructs on media lacking uracil. Moreover, in WT cells, a strong decrease of fusion protein was observed 30 min following a shift to 27 °C (Fig. 6b). Thus, the lt-degron allows a significantly lower restrictive temperature by about 10 °C compared with the classical temperature-controlled $DHFR^{P67L}$ variant.

### Conditional protein depletion in *Drosophila*.

We then tested the lt-degron in *D. melanogaster* using *K2:GFP* and *K2:TEV* as model POIs. *K2:GFP* was destabilized after a shift from stabilizing 15 to 29 °C in embryonic Kc cells subjected to a temperature shift for 4 h (Fig. 7a,b). *K2:TEV* was depleted from macrophage-like Schneider 2 (S2) cells after a shift from 16 to 29 °C (Fig. 7c). A stable transgenic *Drosophila* line expressing *K2:TEV* under the control of the *Actin5c* promoter proved the principle *in vivo*. After shifting the living flies from the permissive (18 °C) to the restrictive (29 °C) temperature, we observed that *K2:TEV* becomes instable (Fig. 7d). This revealed that the lt-degron has a broad spectrum of applications and can be used to modulate protein abundance not only in plants and yeast but also in animal cell culture and living insects.

## Discussion

Tight modulation of protein activities is still a major challenge, especially in multicellular organisms. We introduce here a transferable lt-degron for obtaining conditional mutants coupled to tunable protein accumulation in entire plants, insects and cell culture. The lt-system allows a rapid and reversible modulation of protein function and enzymatic activities. This was demonstrated in the case of seven different targets in five model systems, suggesting that it is generic and versatile.

Pronounced and unique technical advantages of the method over comparable systems include non-invasive functionality in a wide range of multicellular organisms, the lack of technically difficult or costly applications of exogenous chemical triggers, the possibility of the formation of cell and tissue types on demand by direct action on protein activity, and the perspective of applicability in large scale, for example, in entire growth containments, as the trigger is temperature.

In today's age of recent technological advances where site-specific gene inactivation is becoming routine, clearly methods for the re-introduction of tightly and rapidly controllable conditional genes will allow additional functional analyses. In combination with CRISPR/Cas9, ZFNs or TALENs, our method allows directed generation of novel conditional alleles of genes, including essential ones, at the organismal and tissue-specific level. The lt-degron is technically well suited for the application in poikilothermal animals such as fishes, amphibians, reptiles, insects and other invertebrates, facilitating the design of conditional knockout animals in many model systems to dissect, for example, developmental programmes.

We recently discussed the classical heat-inducible N-degron with other techniques for conditionally altering protein abundance and generating phenotypes on demand (Supplementary

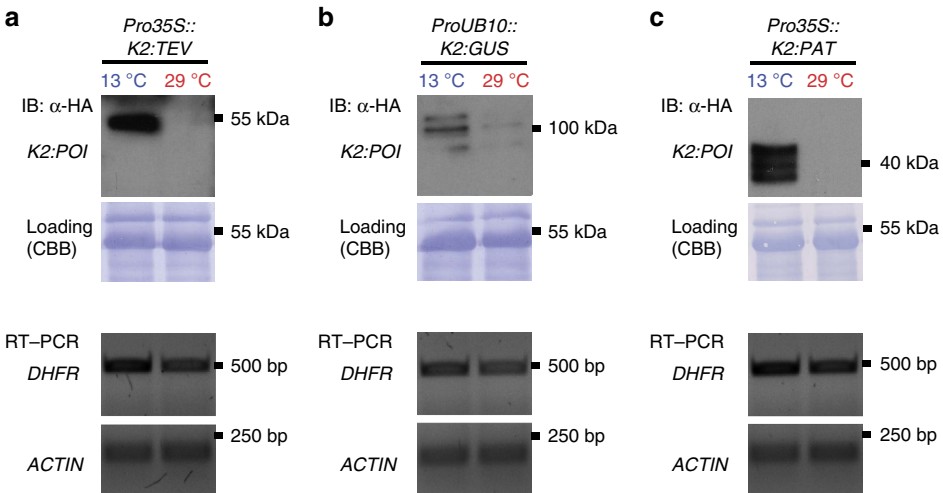

**Figure 5 | The lt-degron in *N. benthamiana* as a protein accumulation system.** (**a**) Stability and transcript levels of *K2:TEV* in transiently transformed *N. benthamiana* (tobacco) plants. (**b**) Conditional expression of *K2:GUS* and (**c**) of *K2:PAT*. Equal loading was confirmed by staining of blotted membranes with Coomassie Brilliant Blue G225 after immunostaining.

Note 1)[22]. In contrast, the system presented here is expected to work for many proteins under physiological conditions in multicellular, intact organisms, however, different aspects have to be taken into consideration when adopting this technique: (1) POIs must tolerate N-terminal fusions of about 26 kDa, a C-terminal fusion is not feasible; (2) POIs must at least

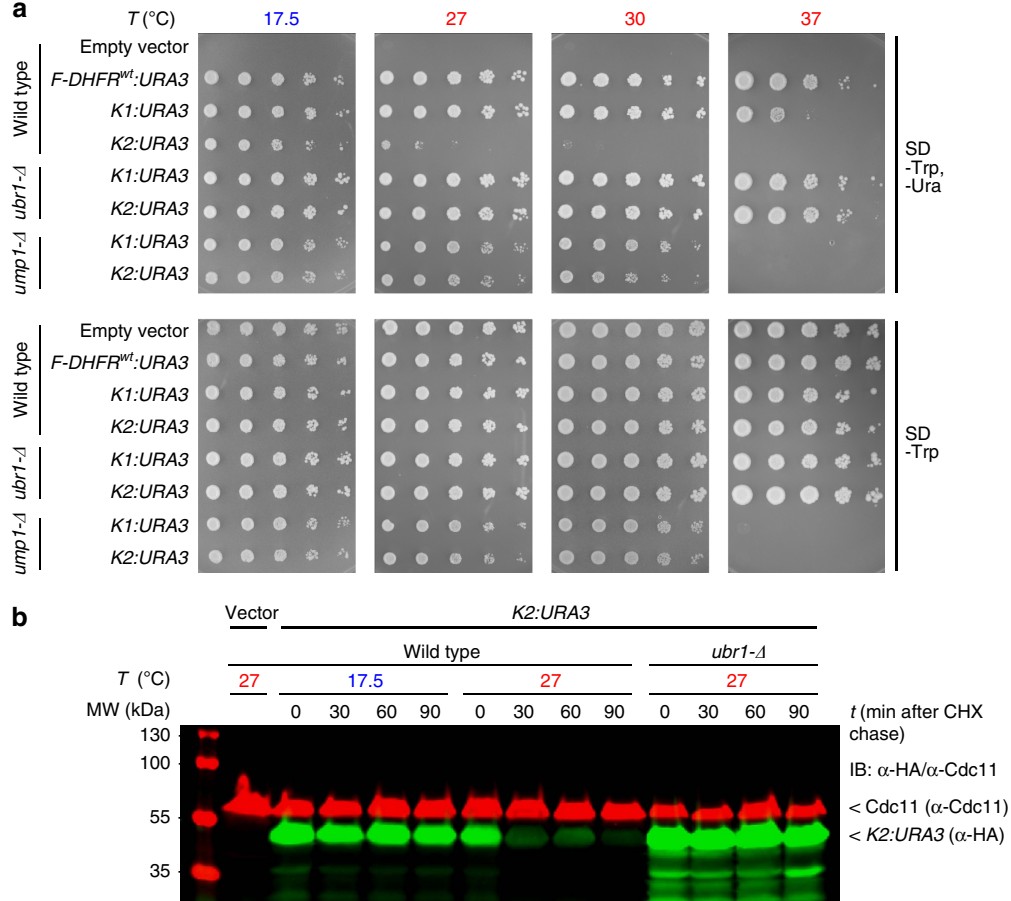

**Figure 6 | The lt-degron significantly lowers the restrictive temperature for conditional alleles in *S. cerevisiae*.** (**a,b**) Modulation of enzyme activity and protein degradation in *S. cerevisiae*. Yeast cells were transformed with *F-DHFR^wt:URA3, K1:URA3, K2:URA3* under control of the copper-inducible P_CUP1 promoter. (**a**) Spot tests of WT, *ubr1-Δ* or *ump1-Δ* cells in serial (1:5) dilutions grown for 3 days at the indicated temperatures. Plates were grown at 17.5 °C for 5 days. *ump1-Δ* shows a general growth defect at elevated temperature. (**b**) Cycloheximide (CHX) chase of *K2:URA3*. Equal loading was confirmed by simultaneously probing against Cdc11 (red signal).

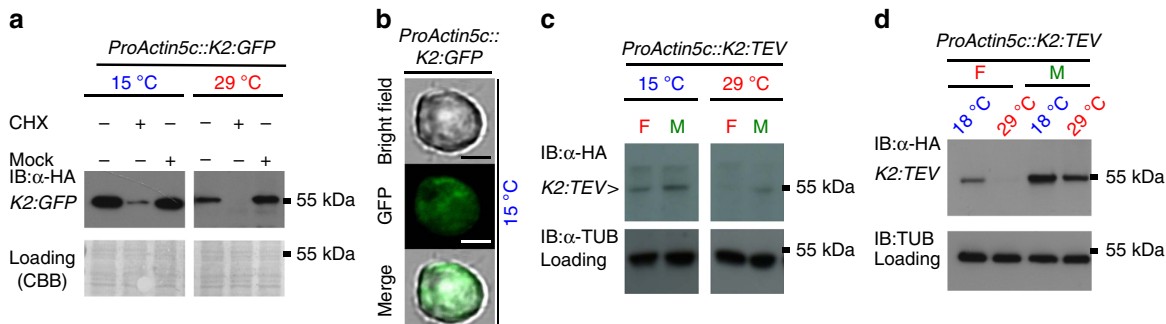

**Figure 7 | Conditional protein depletion in cell culture and living *Drosophila* flies.** (**a–d**) Modulation of protein abundance in *D. melanogaster*. Two of the most commonly used *Drosophila* cell lines, as well as stably transformed living flies were used as follows. (**a**) *K2:GFP* stability in embryonic *Drosophila* Kc cells with 24-h recovery after transfection and after a temperature shift for 4 h from permissive (15 °C) to restrictive temperature (29 °C). CHX chase was performed with 100 μg ml⁻¹ in DMSO. (**b**) *K2:GFP* functionality detected as green fluorescence at permissive temperature. Scale bar, 2 μm. (**c,d**) Depletion of *K2:TEV* depending on a destabilizing N-terminal residue (F: phenylalanine) according to the N-end rule. Methionine (M) as control. (**c**) Transfection into *Drosophila* Schneider 2 (S2) cells with 24-h recovery after transfection and 60-h post-transfection temperature shift from 16 to 29 °C. (**d**) *Drosophila* flies were stably transformed with *K2:TEV* initiated by a Phe (F) residue or a *M-K2:TEV* control. Flies were subjected to shifts to permissive (18 °C) and restrictive temperatures (29 °C). Equal loading was further confirmed by staining of blotted membranes with probing against tubulin. CBB, Coomassie Brilliant Blue.

temporally localize to the nucleus or cytosol, the compartments with high activity of the N-end rule pathway or—in case for membrane localized proteins—have accessible N termini[13]; (3) a pre-selection via phenotype and/or western blotting after transformation helps to identify lines responding to the temperature alterations. The functionality of the system in organelles other than the cytosol or the nucleus where the action of the N-end rule pathway is unclear such as endoplasmic reticulum and other membrane systems, mitochondria and chloroplasts is still to be determined.

Plant growth and development are known to be influenced by temperature[18,41–44] and these topics are in the focus of current research. Detrimental side effects of temperature treatment may represent a downside of a temperature-induced approach, the impact of 29 °C as a restrictive temperature on plant growth was documented (Figs 1 and 2, and Supplementary Figs 3 and 6). Effects of the plant high-temperature adaptation responses such as hypocotyl and petiole elongation, increased leaf size and earlier flowering have to be taken into consideration. Effects of elevated temperature on plant development and genetic control of plant thermomorphogenesis are currently matter of intense research[45]. However, all tested organisms were able to degrade the K2 fusion proteins at restrictive temperature, and transcript levels remained stable over permissive and restrictive conditions. This suggests a low vulnerability of our lt-degron system to side effects related to growth differences caused by temperature.

In this study, the lt-degron enabled us to determine the temporal requirement of the transcriptional regulator TTG1 in the context of trichome development. With a switchable CO transcription factor, we controlled timing of flowering and generated specialized cell types and meristematic tissue of the flower on demand. An engineered conditional TEV protease offers opportunities for, for example, a facilitated downstreaming of recombinant proteins by intracellular self-cleavage directly in the expression host. K2:GUS temperature-shift experiments coupled to GUS activity measurements revealed that the lt-degron opens completely new avenues by, for example, testing the influence of various levels or pulses of conditionally accumulated proteins of interest in one stable genetic background without the need to generate a plethora of transgenic events.

Trichomes are very well-studied specialized model cells[29] present on the surfaces of many plants and capable of synthesizing and either storing or secreting large amounts of specialized metabolites[30]. The use of the intrinsic metabolic capacity of trichomes to produce secondary metabolites or to enhance transcript levels of genes encoding enzymes required for biosynthesis of small molecules within trichomes has been discussed[46–49].

A special feature of using the heat-inducible N-degron system in living organisms is that it allows conditional build-up of this cellular trichome compartment in the K2:TTG1 system. This could be used as a genetic background to globally switch ON/OFF a regulatory protein and establish a specific cellular reaction compartment or micro reactor. Such a cell population or tissue could harbour an enzymatic reaction cascade needed to build-up a molecule of interest, which is not necessarily a protein, which could potentially directly be influenced by the fusion to the lt-degron. Compartment-specific expression can be achieved by using compartment-specific promoters. Thus, the manufacture of biomolecules represents a possible scalable technical application as a production platform in the context of synthetic biology. We are convinced that the lt-degron is a powerful tool both for fundamental research in cell biology and developmental genetics but also has strong potential in biotechnological contexts like molecular farming of proteins or for metabolic engineering of small molecules. The conditional formation of trichomes in a

trichome-lacking mutant or induced development of flowering tissue can be regarded as the design of a reaction compartment on demand, a tool to possibly form *in vivo* bioreactors.

## Methods

**Plant work.** *A. thaliana* (L.) Heynh. plants were sown on soil mixture 1 (eight bags of MiniTray (70 l per bag, Balster Einheitserdewerk); add 50 l of water containing 800 ml Osmocote Start (Scotts International) and 250 g BioMükk (Sautter and Stepper)), or on steamed (pasteurized for at least 3 h at 90 °C) soil mixture 2 (Einheitserde Classic Kokos (45% white peat, 20% clay, 15% block peat, 20% coco fibres; 10-00800-40, Einheitserdewerke Patzer); 25% Vermiculite (grain size 2–3 mm; 29.060220, Gärtnereibedarf Kamlott) and 300–400 g Exemptor per m³ soil substrate (100 g kg⁻¹ thiacloprid, 802288, Hermann Meyer). After stratification of 4 to 5 days at 4 °C in the dark, seeds were germinated and plants grown under standard long-day (16/8 h light/dark) or short-day (8/16 h light/dark) greenhouse conditions between 18 and 25 °C. *N. benthamiana* (tobacco) plants were grown on soil mixture 2.

For strictly controlled development such as during temperature-shift experiments, plants were grown either in growth cabinets (AR-66L2 and AR-66L3, Percival Scientific, CLF PlantClimatics) or in walk-in phyto chambers (Johnson Controls, equipped with ESC 300 software interface) at a humidity of 60% depending on the requirements, and watered with pre-warmed or pre-cooled tap water.

Seeds were plated either on soil or aseptically *in vitro* on plastic Petri dishes under long-day regime (16/8 h light/dark) on 0.5% Murashige & Skoog (MS; Duchefa Biochemicals, M0221) containing 1% sucrose and 8 g l⁻¹ phytoagar (Duchefa Biochemicals, P10031) in growth cabinets. For aseptic culture, dry seeds were sterilized with chlorine dioxide gas produced from 75% *Eau de Javel* (Floreal Haagen) and 25% HCl. Selective MS media contained 10 mg l⁻¹ DL-phosphinotricin (PPT, Basta, glufosinate-ammonium, Duchefa Biochemicals), 50 mg l⁻¹ kanamycin sulfate, 5.25 mg l⁻¹ sulfadiazine sodium salt (Sigma-Aldrich, S6387) or 20 mg l⁻¹ hygromycine B (Duchefa Biochemicals, H0192). Basta-resistant plants were selected *in solium* in cotyledon stage by spraying 150 ml per tray of a 1:1,000 dilution of Basta (contains 200 g l⁻¹ glufosinate-ammonium; Bayer CropScience) in tap water, which was repeated three times in a 2-day interval.

Plants used in this study were all in the background of the Columbia-0 accession (Col-0) and either WT plants, T-DNA insertion mutants for *TTG1* (*ttg1*, GABI_580A05, NASC stock ID: N455589, kindly provided by the GABI-Kat[50] team via NASC—The European Arabidopsis Stock Centre, arabidopsis.info), ethyl methanesulfonate mutants for *PRT1* (*prt1* HygS or *prt1-1*, kind gift of Andreas Bachmair, Max F. Perutz Laboratories, Vienna, Austria)[24–26] and 'TTG1-HA' is *ttg1-13 Pro35S::TTG1-3xHA* (kind gift of Martina Pesch, University of Cologne, Germany. *ttg1-13* is a deletion line from fast-neutron bombardment, originally isolated from David Oppenheimer, University of Alabama, Tuscaloosa[31].), which carries the *TTG1* CDS in the backbone of *pAM-PAT-GW-3xHA* (see below for plasmid details) under control of the *CaMV 35S* promotor (*Pro35S*) and carrying the *bar* resistance marker (Laurent D. Noël and Jane Parker, Max Planck Institute for Plant Breeding Research, Cologne, Germany)), which was used as a negative control for a temperature-dependent destabilization of TTG1. Leaves 3/4 of the *ttg1* mutant show few undifferentiated stichel-like trichomes or initiation sites at the rim of the leaf, but neither initiation sites proximal to the midvein nor further differentiated trichomes (Figs 1c and 2a). The central parts remain glabrous[31]. All plant strains used are listed in Supplementary Table 3.

For cryo scanning electron micrographs (SEMs), 5- to 7-day-old plants (pulses; Fig. 2b or 2-week-old plants (shifts; Fig. 2c) were treated for the indicated time. First, leaves were not yet visible at the time of the pulse experiments. Before protein isolation, transgenic plants were subjected to temperature shifts by removing the potted plants constantly grown at the indicated starting temperature (16 °C in Fig. 2b or 29 °C in Fig. 2c) and shifting them to the corresponding destination temperature (29 °C in Fig. 2b or 16 °C in Fig. 2c). To achieve a rapid acclimatization, plants were immediately watered with water pre-warmed or pre-cooled to the final temperature.

**DNA work and degron construct design.** DNA cloning was performed following standard procedures using *Escherichia coli* strain DH5α (Invitrogen). The constructs were generated by fusion PCR (*Pfu* polymerase, fermentas) using the primer combinations according to Supplementary Table 4. All individual constructs designed for this study are schematically represented in Supplementary Fig. 2. DNA amplicons were purified with ExoSap-IT (USB) preceding fusion PCRs. All fusions were flanked by Gateway *attB1/attB2* sites and recombined by BP reactions into *pDONR201* (Invitrogen).

Constructs *K1–K3* start with the strong *CaMV 35S* promoter, a 3′-tobacco Omega (Ω) leader sequence for translation enhancing followed by a sequence encoding for a synthetic codon-optimized *Ub*, a triplet for the destabilizing bulky and hydrophobic amino acid Phe (F), a short linker peptide (e^L; translates to HGSGI)[34] to further expose the N-terminal residue, one of the three different temperature-sensitive mouse dihydrofolate reductase sequences (*DHFR^ts*) triggering the protein unfolding response, a triple hemagglutinin (HA)-tag (HAT)

for immunodetection and the gene of interest. All plant degron cassettes carry Phe as destabilizing N-terminal residue, because it was shown to be recognized by the *Arabidopsis* N-end rule pathway[24–26] but to also represent a strongly destabilizing N-terminal in other model systems[6–9,25,26] (Fig. 1a,b and Supplementary Fig. 1).

Cassettes *K1–K3* were based on a 5′-synthetic human *Ub* gene[51] and an Ω leader sequence from *pRTUB8*, a derivative of *pRTUB1* (refs 24,52). The leader contains the 20 nucleotides upstream of the start codon of tobacco mosaic virus strain U1 (ref. 53) HAT was amplified from *pSKTag3SUM6* (kind gift of Andreas Bachmair). The differences of the three *DHFR* variants in *K1–K3* are as follows.

For construct *K1*, the yeast N-degron cassette, the mouse *DHFR$^{P67L}$* variant[1,23], was amplified from *pKL187* of the *S. cerevisiae* 'DEGRON-KIT' ('Kit for making *CUP1* t.s. degrons by PCR', EUROSCARF, EUROpean *Saccharomyces cerevisiae* ARchive for Functional Analysis (web.uni-frankfurt.de/fb15/mikro/euroscarf/))[54,55]. Ω leader and Ub were amplified from *pRTUB8* with ND70/ND71, *DHFR$^{P67L}$* from *pKL187* with ND72/ND73 and HAT from *pSKTag3SUM6* with ND74/N138 (*K1:CO*) or N172 (*K1:TTG1*) introducing a *NotI* site. The short linker e$^L$ (translates to HGSGI) was adapted from previous studies on protein half-life[34] and used to further expose the N-terminal Phe. The TTC triplet introducing Phe (F) at the *Ub–DHFR* junction and the sequence encoding e$^L$ were included in ND71/ND72. Final fusion PCRs were always accomplished using the most distal primer pairs. Cloning and sequencing primers are listed in Supplementary Table 4 and Supplementary Fig. 2.

*K2* is derived from *pJH10$^{mut}$* (*pJH23*), containing a mutated mouse *DHFR$^{T39A,E173D}$*, which was isolated in the yeast mutagenesis screen (Supplementary Note 2)[33]. The fusion partner fragments were generated analogous to construct *K1*, that is, Ω leader and Ub were amplified with ND70/ND78, *DHFR$^{T39A,E173D}$* from *pJH10$^{mut}$* with ND79/ND80 and HAT with ND81/ND75 (*K1:TTG1*) or N138 (*K1:CO*). A point mutation within the *DHFR$^{T39A,E173D}$* was corrected by site-directed mutagenesis using N158/N159 and PfuTurbo polymerase (Stratagene). *TTG1* of *K2:TTG1* contains two silent point mutations (A555G and C567T).

An Entry clone containing *K2:TTG1* was used as a template for further *K2:POIs*. A Gateway-compatible Entry clone *pEN-L1-K2-L2* containing the K2 degron cassette including Ω to HAT was generated to construct *K2:POI* fusions by Gateway LR reactions. *K2* was amplified from the previously described *K2:PAT* using primers 21/22 containing *attB1/attB2* recombination sites.

*K3* contains a triple-mutated mouse *DHFR*. All degron constructs containing cassette *K3* are derived from the Entry vectors harbouring *K2:TTG1/CO*, which were used as templates in site-directed mutagenesis (PfuTurbo, Stratagene) using ND68/ND69 introducing the additional mutation P67L. *DHFR* of *K3* carries the two newly isolated mutations *DHFR$^{T39A,E173D}$* plus *P67L*, the original mutation from the *S. cerevisiae* N-degron[1].

*K4* starts with the native DHFR and an N-terminal Met residue and is based on *pJH10$^{mut}$* (*DHFR$^{T39A,E173D}$*) from the yeast screen (Supplementary Note 2)[33]. *DHFR* was amplified with ND82/ND80 and HAT with ND81/N138 (*K4:CO*) or N172 (*K4:TTG1*). *K2* and *K3* lacking the N-terminal HGSGI linker following the destabilizing Phe residue were based on the corresponding *K2* and *K3* fusions with TTG1 and CO. To alter the *Ub-Phe–DHFR* junction, TR01 and TR02 were designed for fusion PCR.

**Plant expression constructs.** *Arabidopsis TTG1* and *CO* cDNAs (AT5G24520 and AT5G15840; kind gifts of Daniel Buoyer, University of Cologne, and Laurent Corbesier, Max Planck Institute for Plant Breeding Research, Cologne) were amplified with N173/ND77 (*K1:TTG1* and *K4:TTG1*), with ND76/ND77 (*K2:TTG1* and *K3:TTG1*) and N139/N140 (*K1:CO*, *K2:CO*, *K3:CO* and *K4:CO*).

These *degron:POI* fusion proteins were subcloned into *pDONR201* (Invitrogen) and the Entry clones sequenced (Supplementary Table 5). For stable expression in transgenic plants, Entry clones were recombined in an LR reaction into the *attR* site-containing binary Gateway destination vector *pLEELA* (ref. 56), containing a double *CaMV* 35S promoter fused to the first *WRYKY33* intron or into *pAM-PAT-GW-ProUBQ10* (generated and kindly provided by Stefan Pusch, Deutsches Krebsforschungszentrum, Heidelberg). These Gateway-compatible Destination vectors are derived from *pAM-PAT-MCS* (multiple cloning site; GenBank accession number AY436765), derivative of *pPAM* (GenBank accession number AY027531)[57]. These backbones carry the *Streptomyces hygroscopicus bar* gene that translates to phosphinothricin-*N*-acetyltransferase (PAT) as plant selection marker conferring resistance towards PPT. The vector backbone of *pAM-PAT-GW-ProUBQ10* contains an insertion of 132 bp between the UBQ10 promoter and the attR1 site. The insertion is part of the promoter of *TOO MANY MOUTHS* (AT1G80080) and located after the 5′-XhoI site used to insert the UBQ10 promoter into the *pAM-PAT-GW* backbone. The insertion does not result in alternative translation products since it includes stop codons in all reading frames.

For *K2:GFP*, *pEN-L1-K2-L2* was recombined with *pAM-Kan-35S-GW-GFP* (kind gift of Jane Parker), also a derivative of *pAM-PAT-MCS*.

*K2:GUS* was assembled by amplifying *K2* with ND70/TR08 from *K2:TTG1*, as well as by TR07/TR06, which amplifies *GUS* from *pENTR-gus* (*uidA* CDS[58] in *pDONR201* (carries a Val(GtC)2Leu(TtA) and a silent Gaa>Caa substitution for Glu279)). *K2:GUS* was fused using ND70/TR06 and introduced into *pAM-PAT GW ProUBQ10* by LR.

For *K2:TEV*, *K2* was amplified from *K2:TTG1* with primers 41/36. *TEV* was isolated from *pCT190-6* (ref. 19) with primers 46/29, generating linker sequence 'L4' comprising HAT and hexahistidine tags. *pCT190-6* carries a TEV-containing

fusion protein in *pRS416*, TEV is a more stable, autoinhibition-resistant mutated (S219V) catalytic domain variant[59]. *K2* and *TEV* were fused with primers 41/29, and *Taq* was added for a final 15-min adenylation step at 72 °C. The *K2:TEV* product was incompatible with Gateway ENTR vectors suggesting that the TEV sequence interferes with plasmid stability. Thus, *K2:TEV* was cloned into *pCRII-TOPO* (Invitrogen) by TOPO cloning between the XhoI and SpeI. To construct a binary plant expression vector, *K2:TEV* was then cloned into *pAM-PAT-MCS* using the 5′- and 3′-EcoRI sites from *pCRII-TOPO*.

*K2* for *K2:PAT* was amplified from *K2:TTG1* with ND70/TR11, *bar* (encodes PAT) from *pLEELA* with TR10/TR09, and fused with ND70/TR09. To receive a kanamycin-selectable binary plant expression vector, *K2:PAT* was recombined into *pJan33* (double *CaMV* 35S promoter fused to the first *WRYKY33* intron, selectable marker *NPTII*; kind gift of Marc Jakoby, University of Cologne).

**S. cerevisiae constructs.** Plasmids are derivatives of *pRS314* (*CEN-ARS-TRP1*) or *pRS315* (pJH10, *CEN-ARS-LEU2*)[60]. DHFR-HA-URA3 was assembled by inserting copper-inducible promoter P$_{CUP1}$ as a NotI–EcoRI fragment, *Ub-F-DHFR* as EcoRI–BamHI fragment from *pJH10* (WT DHFR), *DHFR$^{P67L}$* (same as in K1), *pJH10$^{mut}$C33* containing the *DHFR$^{T39A,E173D}$* variant (same as in K2; Supplementary Note 2) and HA-URA3 as BamHI–KpnI fragment into the polylinker of *pRS314* serving as templates for *pJD646* (*Ub-F-DHFR-wt*), *pJD647* (*Ub-F-DHFR-P67L = K1*) and *pJD648* (*Ub-F-DHFR-T39A, E173D = K2*). Yeast strains used are listed in Supplementary Table 3. Standard media were YPD medium and ammonia-based synthetic complete (SC) dextrose.

For spot growth assays, yeast cultures were grown overnight in liquid culture and exponentially growing cells serially diluted to 5,000/1,000/200/40/8 cells per µl (1:5 steps). A volume of 1 µl of the suspension was spotted with a pin-frogger onto plates containing yeast standard minimal medium/synthetic defined (SD) and grown for 3–5 days.

For cycloheximide (CHX) chase experiments, exponentially growing yeast cultures were supplemented with 100 µg ml$^{-1}$ CHX; samples collected at different time-points, pelleted, washed and frozen in liquid N$_2$. The cells were lysed by incubating them in Laemmli loading buffer at 100 °C for 5 min. Extracts were analysed by SDS–polyacrylamide gel electrophoresis (SDS–PAGE) and two-colour western blotting using the Odyssey Infrared Imaging System (Li-Cor). Mouse monoclonal anti-HA epitope tag antibody (HA.11, clone 16B12, MMS-101R, BioLegend) in combination with an IRDye 800-coupled rat anti-mouse IgG (Rockland) was used for detecting tagged *DHFR-HA-URA3* proteins. A rabbit polyclonal anti-Cdc11 antibody (y-415: sc-7170, Santa Cruz Biotechnology) was used for detecting CDC11 in combination with an Alexa Fluor 680-coupled goat anti-rabbit IgG antibody (Biomol), which served as an internal protein-loading control[61].

**Drosophila constructs.** For expression of *K2:GFP* in *Drosophila* cell culture, *K2* from *pEN-L1-K2-L2* was introduced into *pAWG* (*Drosophila* Genomics Resource Center (DGRC), Indiana University, Bloomington, USA; https://dgrc.cgb.indiana.edu/product/View?product=1072; Stock Number: 1072; Murphy, T.D. *et al.*, Construction and application of a set of Gateway vectors for expression of tagged proteins in *Drosophila*, unpublished data) via a Gateway LR reaction. *pAWG* contains an *Actin5c* promoter upstream of the Gateway cassette and a C-terminal *eGFP* tag. *ProActin5c::K2:TEV* for cell culture transfection and transformation of flies contains *DHFR$^{T39A,E173D}$*, amplified from *K2:TTG1* using primers 36/37 (*Phe-K2*) or 36/38 (*Met-K2*). Ub from *S. cerevisiae*[4] was used and amplified with primers 9/39 (*Phe-K2*) or 9/40 (*Met-K2*). The fragments were fused to *Ub:X-DHFR$^{T39A,E173D}$* with primers 9/36. TEV was amplified from *pCT190-6* with primers 46/29 and *K2* fused with primers 9/29. *Taq* polymerase was added for a final 15-min adenylation step at 72 °C. The construct was subcloned in *pCRII-TOPO*. *K2:TEV* was incompatible with Gateway recombination and the construct prepared as XhoI–SpeI fragment, which was partially cut to leave an internal SpeI site unaffected. The fragment was ligated into *pAW* (DGRC; https://dgrc.cgb.indiana.edu/product/View?product=1127; Stock Number: 1127). *pAW* contains the *Actin5c* promoter upstream of the Gateway cassette and was cut with XhoI and NheI to lose the Gateway cassette and yield the *Actin5c*-driven fly expression vector.

**Stable plant transformation and selection of transformants.** The binary plant expression vectors were retransformed into *Agrobacterium tumefaciens* GV3101-pMP90RK (C58C1 Rif$^r$ Gm$^r$ Km$^r$)[62] to obtain a bacterial transformation suspension. The identity of the *Agrobacterium* strains was verified by backtransformation of isolated plasmid into *E. coli* DH5α and at least three independent analytical digestions. All constructs were transformed by a modified version of the floral dip method[63].

Individual T1 (generation 1 after transformation) transgenic plant lines were pre-selected with Basta or kanamycin as described above. To exclude lines showing position effects, for example, by disrupting essential genes by the construct T-DNA, the number of insertion loci was determined in a segregation analysis in the T2 generation and only transgenic plants carrying one single insertion locus were further used. Standard lines were established by isolating T3 plants homozygous for the transgene. Independent representative reference lines displaying a typical conditional phenotype on temperature up- and downshifts were used in the final

experiments. Standard lines were established by isolating T3 plants homozygous for the transgene.

To identify responsive transgenic lines, we prescreened *TTG1-td*, *CO-td*, *GUS-td* and *PAT-td* by developmental and histological phenotype and *TEV-td* and *GFP-td* by western blotting.

**Transient transfection of *N. benthamiana*.** For transient transformation of tobacco, leaves of 4-week-old plants were infiltrated with *Agrobacteria* GV3101-pMP90RK, carrying binary plant expression vectors. *Agrobacteria* were grown to the stationary phase overnight in 10 ml of YEB medium and pelleted at 5,000*g* at room temperature. The pellet was washed once in 10 ml of infiltration buffer (10 mM MES (pH 5.6), 10 mM MgSO$_4$, 100 $\mu$M acetosyringone (D134406, Sigma)) and subsequently resuspended in 10 ml of the same buffer. *Agrobacteria* strains were co-infiltrated with *p19*, an *A. tumefaciens* strain based on GV3101 carrying pBIN6Ip19 containing the *p19* protein of tomato bushy stunt virus to suppress post-transcriptional gene silencing and increase ectopic gene expression[64]. Before infiltration, bacteria suspensions were adjusted to an OD$_{600}$ of 0.5. Desired bacteria suspensions were then mixed with *p19* bacteria suspension to a final OD$_{600}$ value of 0.25 for both components. Bacteria suspensions were then infiltrated into the epidermis on the lower side of the tobacco leaf. Infiltrated areas were marked on the upper side of the leaf with a permanent marker. For an easier infiltration procedure, plants were watered and transferred to standard greenhouse long-day conditions the day before to allow plant stomata to open. To allow efficient transformation and expression, plants were kept for 48 h in the greenhouse, before applying temperature shift experiments by transferring them into a growth cabinet at either permissive or restrictive growth conditions. For one data point, 15–20 leaf discs of 5 mm diameter were collected from infiltrated areas and snap-frozen in liquid nitrogen. Extraction was performed using radioimmunoprecipitation assay (RIPA) buffer as mentioned below.

**Drosophila cell culture work.** *D. melanogaster* Kc embryonic tissue culture cells[65] or Schneider 2 (S2) cells[66] (Invitrogen) were grown at 24 °C in 3 ml of Schneider's medium (Gibco) supplemented with 10% fetal bovine serum. Cells were passaged into fresh media every 5 days in a 1:10 dilution. Transfection was carried out in a 12-well dish using the Effectene transfection kit (Qiagen) according to the manufacturer's protocol for adherent cells. Temperature treatments and shifts from 24 °C to the desired temperature were carried out at 15 or 16 °C versus 29 °C. The details of the temperature-shift experiments are indicated in the figure legend. CHX chase was performed with 100 $\mu$g ml$^{-1}$ in dimethylsulfoxide (DMSO).

**Drosophila transformation and characterization.** Vectors carrying *Actin5c::Phe-K2:TEV* (F-K2:TEV) or *Actin5c::Met-K2:TEV* (M-K2:TEV) were injected in ZH-attP-2A or ZH-attP-51D embryos according to the phiC31 recombinase transgenesis system[67]. In brief, 1-h old embryos were collected, placed on slides and immersed in injection oil. The plasmids were injected in the posterior region of the embryo aiming at the area where the pole cells would develop. Hatched larvae were collected and placed on standard *Drosophila* medium based on a mixture of cornmeal, yeast, agar, sucrose, soy flour and water. Transformants were grown at 25 °C standard conditions. Adult flies were shifted either to 18 or 29 °C. After 24 h, total protein extracts were obtained from whole flies homogenized in standard lysis buffer (20 mM Tris-Cl (pH 7.5), 200 mM NaCl, 2 mM EDTA, 10% glycerol, 1% Nonidet P-40 and EDTA-free Complete Protease Inhibitor Cocktail (Roche Diagnostics)).

**Protein extraction and western blot analysis.** Plant tissue (*Arabidopsis*: leafs or seedlings, tobacco: leaf discs) was collected in a standard 2 ml reaction tube containing three Nirosta stainless steel beads (3.175 mm; 75306, Mühlmeier), snap-frozen in liquid N$_2$ and stored at −80 °C. Material was ground frozen using a bead mill (Retsch, MM400; 45 s, 30 Hz) in collection microtube blocks (adaptor set from TissueLyser II, 69984, Qiagen).

For *K2:TTG1* time courses, per time point, one leaf was ground in 200 $\mu$l of extraction buffer (50 mM Tris-Cl (pH 7.6), 150 mM NaCl, 5 mM EDTA, 0.1% SDS, 0.1% ß-mercaptoethanol and EDTA-free Complete Protease Inhibitor Cocktail). Alternatively, tissue was lysed using RIPA buffer (50 mM Tris-Cl (pH 8), 120 mM NaCl, 20 mM NaF, 1 mM EDTA, 6 mM EGTA, 1 mM benzamidine hydrochloride, 15 mM Na$_4$P$_2$O$_7$ and 1% Nonidet P-40 supplemented with EDTA-free Complete Protease Inhibitor cocktail added freshly).

Extraction of *K2:TEV* was done in 25 mM Tris-Cl (pH 7.5), 75 mM NaCl, 15 mM MgCl$_2$, 15 mM EGTA (pH 8.0), 0.1% Tween 20, 0.1% Triton X-100, 5 mM dithiothreitol and EDTA-free Complete Protease Inhibitor Cocktail in a chilled cooling block at 4 °C, 800 r.p.m. for 30 min. Insoluble cell debris was pelleted via centrifugation for 20 min at 4 °C, 20,000*g*. The protein content of the samples was determined using a DirectDetect infrared spectrophotometer (MerckMillipore).

*Drosophila* cells were transferred into a standard 1.5 ml reaction tube and collected via centrifugation at 4 °C for 5 min, 300*g*. The pellet was washed once with ice-cold PBS and cells lysed using RIPA buffer. Following steps were carried out as described above.

*Drosophila* pupae were lysed as described above and western blots performed with rabbit polyclonal anti-HA epitope tag antibody (Y-11: sc-805, Santa Cruz

Biotechnology) or rabbit polyclonal $\alpha$-Tubulin antibody (H-300: sc-5546, Santa Cruz Biotechnology) and detected by enhanced chemiluminescence (ECL).

*S. cerevisiae* cells were grown at 30 °C to logarithmic growth phase, and CHX added to the cultures to a final concentration of 100 mg l$^{-1}$. Strains were expanded at 25 °C and incubated at 37 °C for 30 min before CHX addition. Proteins were extracted from 5 to 50 ml of exponentially growing suspension culture (OD$_{600}$ = 0.8–1.2) by centrifugation and grinding with glass beads (5 min, IKAVibrax-VXR, Janke und Kunkel) at 4 °C in 150–300 $\mu$l Native lysis buffer (50 mM Na-HEPES (pH 7.5); 150 mM NaCl, 5 mM EDTA, 1% Triton X-100, containing EDTA-free Complete Protease Inhibitor Cocktail, 20 $\mu$M MG132) followed by a second centrifugation (4 °C, 10 min, 20,000*g*). Proteins were separated by SDS–PAGE and analysed by quantitative western blotting[61] using an LAS1000 system (Fuji).

Equal protein amounts were resolved by 10% (GUS) or 12% (all others) SDS–PAGE. Blotting of TTG1 and CO samples was done on nitrocellulose transfer membrane by wet blot in a tank blotter (SCIE-PLAS EB10)[63]. All other samples were blotted onto polyvinylidene fluoride transfer membrane by semi-dry blot using a Trans-Blot SD semi-dry electrophoretic transfer cell (170-3940, Bio-Rad).

Equal loading and general protein abundance was confirmed by staining of the blotted and probed membranes after immunostaining with Ponceau S (sodium salt) or Coomassie Brilliant Blue G250 (CBB G150). All antibodies used including parameters are listed in Supplementary Table 6. All degron constructs can be detected with mouse monoclonal anti-DHFR antibody (A-4: sc-74593, Santa Cruz Biotechnology) and mouse monoclonal anti-HA epitope tag antibody (HA.11, clone 16B12: MMS-101R, BioLegend), or rat monoclonal anti-HA epitope tag antibody (High Affinity, clone 3F10: 11 867 423 001, Roche Diagnostics). Horseradish peroxidase-conjugated secondary antibodies (Supplementary Table 6) were detected by ECL using ECL SuperSignal West Pico or Femto (34087 or 34096, Pierce) followed by exposure on autoradiography film. Colorimetric detection of alkaline phosphatase-labelled antibody (Supplementary Table 6) via the nitroblue tetrazolium chloride/5-bromo-4-chloro-3-indolyl phosphate *p*-toluidine (NBT/BCIP) substrate system by tetrazolium precipitation with NBT/BCIP reaction buffer (100 mM Tris-Cl (pH 9.5), 100 mM NaCl, 50 mM MgCl2, 400 $\mu$M NBT (75 mg ml$^{-1}$ stock in 70% DMSO) and 400 $\mu$M BCIP salt (50 mg ml$^{-1}$ stock in DMSO). Staining was done until colour developed and the reaction stopped in 20 mM EDTA in tris-buffered saline (TBS) (pH 8.0) and by washing for three times in water. Western data were confirmed by analysis of at least three biological replicates. Full blot and gel images are provided in Supplementary Fig. 8.

**In situ glucoronidase histology.** Two- to three-week-old plants were collected and submerged in GUS staining solution (1 mM X-Gluc (X-glucuronide sodium salt))[68]. After vacuum infiltration for 30 min, samples were incubated overnight at 37 °C, fixed and cleared in GUS fixation solution (9:1 ethanol/glacial acetic acid for 4 h/room temperature) and stored in 70% ethanol.

**Quantification of glucoronidase activity.** At least five seedlings/sample were collected, pooled and snap-frozen in liquid N$_2$. Proteins from 2-week-old *K2:GUS* seedlings were extracted in GUS extraction buffer (50 mM Na-phosphate, 10 mM EDTA, 0.1% SDS, 0.1% Triton X-100 and freshly added 10 mM β-mercaptoethanol and EDTA-free protease inhibitor cocktail)[68]. A volume of 196 $\mu$l assay solution (1 mM 4-methylumbelliferyl-β-D-glucuronide (4-MUG)) were mixed with 4 $\mu$l of protein extract in a white 96-well plate (Nunc FluoroNunc, Thermo Scientific) for the assay. Fluorometric measurements were done at 37 °C in a spectrophotometer (M1000, Tecan) for 60 min, one data point was taken per minute. Hydrolysis of 4-MUG by GUS generates the aglycone 4-methylumbelliferone (4-MU), which gives a detectable fluorescent emission at 455 nm after excitation at 365 nm. Owing to the online fluorescence measurement of increasing 4-MU, the reaction samples did not have to be separately stopped with 0.2 M Na$_2$CO$_3$ after 60 min as described in the original protocol. Samples were measured for a total of 60 min with one data point taken every minute and normalized to a 4-MU standard. Activity was determined as pmol 4-MU per min and mg of total protein. Control plants expressing *Pro35S::GUS* from the binary plasmid pBI121 (ref. 69) were a kind gift of Diana Schmidt (Leibniz Institute of Plant Biochemistry, Halle).

**Degron tuning assay.** For the time-course K2:GUS experiment, plants were grown aseptically on MS-containing Petri dishes (see above) including 10 mg l$^{-1}$ PPT for 2 weeks at either permissive or restrictive temperatures. A first sample (at least five seedlings) was taken and snap-frozen in liquid N$_2$. Then, the temperature was increased or decreased by 2.8 °C, depending on the design of the time course (warm-cold-warm or cold-warm-cold), and kept for 24 h. Every 24 h, samples were taken before the new temperature change. The time frame allowed plants to run one complete day/night cycle, acclimatize and appropriately express/stabilize protein. The activity was measured as indicated above, all three biological replicates (*n* = 3) consisting of independent plant material were individually measured as three technical replicates.

**Immunoprecipitation for mass spectrometry.** *K2:GUS* plants were grown on plates on 0.5% MS containing PPT at permissive temperature (13 °C) for 2 weeks, collected and frozen in liquid N$_2$. After milling (see above), plant material was lysed

and proteins extracted with extraction buffer containing phosphatase inhibitors and cleared twice by centrifugation (4 °C, 20,000g)[70,71]. Per sample, two microcentrifuge tubes were prepared, each containing 250 µl of lysate at a concentration of 1.5 µg total protein per µl. Preclearing of sepharose matrix (nProtein A Sepharose 4 Fast Flow, GE Healthcare Life Sciences) was performed for 2 h on a rotating wheel at 4 °C with 100 µl of 50% bead slurry/reaction. After a centrifugation step (4 °C, 5 min, 20,000g), the supernatant was transferred into a new tube and 50 µl of anti-DHFR antibody (A-4: sc-74593, Santa Cruz Biotechnology) were added per reaction. Samples were incubated overnight on a rotating wheel at 4 °C. A volume of 250 µl of 50% bead slurry were then added and incubated for 3 h rotating at 4 °C. Beads were centrifuged (4 °C, 400g, 2 min) and washed 3 times with 1 ml of bead buffer[70,71]. Beads were boiled for 15 min in 30 µl of 5 × SDS loading dye, centrifuged (5 min, 20,000 g) and the supernatants of two reactions pooled. A volume of 5 µl of the supernatant were loaded on a 6% PAGE gel and subjected to western blotting with rabbit anti-HA antibody (HA.11, clone 16B12: MMS-101R, BioLegend) to confirm running parameters of the different K2:GUS bands. The rest of the supernatant was separated via SDS–PAGE and silver-stained using an MS compatible staining protocol in an overall volume of 25 ml per gel. The gel was incubated twice for 20 min in aqueous fixing solution (40% methanol and 10% glacial acetic acid), pretreated for 30 min in 0.8 mM $Na_2S_2O_3$, 0.8 M $CH_3COONa$ and 30% methanol, washed for three times in water for 5 min, stained for 20 min in 12 mM $AgNO_3$, developed for maximum 5 min in 0.2 M $NaCO_3$, 0.04% formaldehyde, stopped for 10 min in 0.3 M $Na_2$-EDTA and washed for two times in water for 10 min. The stained gel can be conserved overnight or longer in 25% ethanol, 3% glycerol and 72% water. Three lanes were excised (Supplementary Fig. 7). Liquid chromatography–mass spectrometric (LC–MS) analysis was carried out as follows.

**LC-MS analysis.** Cysteine side chains were reduced and alkylated, proteins digested with trypsin overnight and peptides extracted and dried in a concentrator. Peptides were dissolved in 5% acetonitrile and 0.1% trifluoroacetic acid, and injected into an EASY nano-LCII chromatography system (Thermo Fisher). Reverse phase chromatography was performed on an EASY column SC200 (10 cm, inner diameter: 75 µm, particle size: 3 µm) using a pre-column (EASY column SC001, 2 cm, inner diameter: 100 µm, particle size: 5 µm) and a mobile phase gradient from 5 to 40% acetonitrile, 0.1% trifluoroacetic acid in water in 150 min (flow rate of 300 nl min⁻¹; all from Thermo Fisher). Peptides were electrosprayed on-line into an LTQ-OrbiTrap Velos Pro mass spectrometer (Thermo Fisher; spray voltage of 1.9 kV). Orbitrap (Fourier transform, FT) and ion trap (IT) injection waveforms were enabled, one micro scan acquired for all scan types, FT master scan preview and charge state screening enabled; singly charged ions were rejected in data-dependent acquisition (DDA) analysis. The mass spectrometer was calibrated with m/z of 445.120024 Th. Data-dependent MS/MS spectral acquisition of the 20 most abundant ion signals was performed in the full scan. SwissProt protein sequence database (535,248 sequences; 189,901,164 residues) was searched with MS/MS spectra using Mascot (matrix Science, Proteome Discoverer v.1.4; Thermo Fisher) to identify Ub, DHFR and GUS peptides. The enzyme was set to trypsin, two missed cleavages tolerated, precursor tolerance set to 7 p.p.m., fragment ion tolerance to 0.8 Th, and the family-wise error rate controlled using a 1% peptide false discovery rate.

**Proteasome inhibitor experiments.** Seedlings were grown in liquid culture in 50-ml Erlenmeyer flasks in 20 ml of 0.5% MS including vitamins (Duchefa Biochemicals, M0222), 0.25% sucrose, 500 mg l⁻¹ MES (2-(N-Morpholino)-ethane sulfonic acid, Carl Roth, 4256) and 10 mg l⁻¹ PPT for 2 weeks at permissive or restrictive temperature. Before treatment, seedlings were transferred into 12-well microplates (Greiner) into 5 ml of medium and treated with 5 µM of MG132 (UBPBio) in DMSO (5 µl of a 50 mM stock per 5 ml medium). Mock-treated samples were supplemented with an equivalent volume of DMSO. Plates were incubated for 5 h, collected, frozen in liquid $N_2$ and proteins extracted with RIPA buffer. After protein quantification with bicinchoninic acid assay (Thermo Scientific), 20 µg of total protein were loaded per lane, separated on a 10% SDS–PAGE, blotted and probed as described above.

**RNA work and RT–PCR.** Two-week-old seedlings were collected and snap-frozen. After milling, about 50 mg of tissue was used for RNA extraction with RNeasy Plant Mini Kit (Qiagen). RNA was measured with a spectrophotometer and quality assessed via agarose gel electrophoresis. For first-strand cDNA synthesis, 500 ng of total RNA were used with an equimolar mixture of four oligo(dT) primers (CDSIII-NotIA/C/G/T, Supplementary Table 7) and RevertAid H Minus Reverse Transcriptase (Thermo Scientific). A volume of 1 µl of cDNA was used for PCR analysis using self-made Taq in two reactions per sample; one with generic degron-specific primers (DHFR_frw/DHFR_rev) to test transcript levels of the transgene and one with intron-spanning primers EF1ss/EF1as for ELONGATION FACTOR 1 (EF1) as a housekeeping gene (amplicon sizes: genomic 810 bp; cDNA 709 bp). Both PCRs were run for 30 cycles. Oligonucleotide primers used for RT–PCR are listed in Supplementary Table 7.

**Microscopy.** SEM was done with a SUPRA 40VP (Carl Zeiss MicroImaging) equipped with a K1250X Cryogenic SEM Preparation System (EMITECH), a CPD

030 critical point dryer (Bal-Tec) and a SC 7600 sputter coater (Polaron) at the on-campus microscopy core facility Zentrale Mikroskopie (CeMic) of the Max Planck Institute for Plant Breeding Research, Cologne. Light and confocal laser scanning microscopy was performed with an LSM710 system (Carl Zeiss MicroImaging). K2:GFP fluorescence was observed in root tips of plants that were aseptically grown for 2 weeks under long-day conditions at either constitutively restrictive or permissive temperatures on 0.5 MS medium containing 50 mg l⁻¹ kanamycin sulfate. Photographs of in vitro cultures and histological stainings were taken with a stereo microscope (Stemi 2000-C) equipped with a Zeiss CL 6000 LED illumination unit, and a video adapter 60 C including an AxioCam ERc 5s digital camera (all from Carl Zeiss MicroImaging).

**Data availability.** The authors declare that the data supporting the findings of this study are available within the article and its Supplementary Information files or are available from the corresponding author on request.

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

## Acknowledgements

This work was supported by a grant for a Junior Research Group of the ScienceCampus Halle—Plant-based Bioeconomy to N.D.; by a PhD fellowship of the Landesgraduiertenförderung Sachsen-Anhalt and a grant of the German Academic Exchange Service (DAAD) awarded to F.F.; grants from the EMBO Young Investigator Programme (YIP) and the Action Thématique et Incitative sur Programme (ATIP) from the Centre National de la Recherche Scientifique (CNRS) to A.S. Financial support comes from the state of Saxony Anhalt, the Deutsche Forschungsgemeinschaft (DFG), Graduate Training Center GRK1026 'Conformational Transitions in Macromolecular Interactions' at Halle and the Leibniz Institute of Plant Biochemistry (IPB) at Halle, Germany. The GABI-Kat project (German Plant Genomics Program—Kölner *Arabidopsis* T-DNA lines) provided the *ttg1* T-DNA line via NASC—The European *Arabidopsis* Stock Centre. We thank Dieter Lange for assistance in western blotting, Petra Majovsky for LC–MS analyses, Hugo Stocker for his advice and Angela Baer and Anna Maria Strässle Eugster for technical support during the fly work. We are thankful to Petra Jansen, Sabine Voigt and Philipp Plato for their excellent greenhouse support at Halle. We thank Annika K. Weimer for constant support; Andreas Bachmair for the *prt1* mutant and initial discussions; Luz Irina A. Calderón Villalobos, Moritz K. Nowack, Nikolay Rozhkov, Christof Taxis, Michael Knop, Masato Kanemaki and the members of N.D.'s lab for advice and comments on the manuscript. The publication of this article was funded by the Open Access fund of the Leibniz Association.

## Author contributions

N.D. and A.S. planned the project and N.D. wrote the manuscript and compiled the display items; N.D. designed the degron cassettes, tested and identified the functional lt-degron cassette and performed plant and protein work for TTG1 and CO together with T.R.; F.F. contributed the GUS, TEV and GFP work in *Arabidopsis*; S.M. and

F.F. contributed protein work in tobacco and transcript analyses; N.D. contributed the developmental and protein analysis of TTG1 and CO. K.N. prepared and performed experiments in cell culture together with F.F.; K.N. prepared and performed experiments in *Drosophila* flies together with I.A.; M.S.F., J.H. and R.J.D. contributed mutant DHFR variants; R.J.D. planned and tested the *S. cerevisiae* constructs; W.H. analysed protein samples by LC–MS; W.B. modelled the DHFR variants. All authors discussed the results and contributed in the finalization of the manuscript. Responsibility for figures and related experiments: Figs 1a,b,c,e, 2a,b and 3b, and Supplementary Figs 1, 2, 3b,c,d,f,g, 5 and 7, N.D.; Figs 1f, 3c, 4, 5 and 7a–c, and Supplementary Figs 3e, 4, 6 and 7a,b, F.F.; Figs 1d, 2c and 3a, and Supplementary Figs 3a,b, T.R.; Fig. 5 and Supplementary Figs 3e and 4a–c,e,g, S.M.; Fig. 6, R.J.D., M.S.F., J.H.; Fig. 7d, I.A., K.N.; Supplementary Fig. 5a–c,e–i, W.B.; Supplementary Fig. 7c, W.H.

## Additional information

**Competing financial interests:** The Technology Offices at the Leibniz Institute of Plant Biochemistry (IPB) and at the Centre national de la recherche scientifique (CNRS) have filed a patent application on behalf of N.D. and A.S. based on the findings described in this manuscript. The remaining authors declare no competing financial interests.

