## [Peer Review File · Nature Communications]

Comments to the reviewer for NCOMMS-16-00173-T

General remarks:

- We have included new data as suggested and listed these alterations after the comments to the reviewer below.
- We edited our manuscript according to the format requirements and suggestions and have highlighted all changes in the word file.

Referee comments:

1) In general, the text in the figures (e.g. labels of blots etc) is quite small, and should be increased in size to make them clearer.

Thanks for the note on visibility and clarity of labeling our display items. We were sure that we stuck to the requirements by nature journals, i.e., 10 pt for labels of the individual panels and 6 pt for labels within the panels, images, and graphs. We didn't find any further hints at Author Instructions of Nature Communications and some labels might either way be redone in copyediting, right?

2) Particularly early on in the paper there is a lot of information that has been placed in supplemental, which was often quite confusing. Can I suggest that each separate subsection of e.g. supplemental notes (and later on, supplemental discussion) be numbered, so that in the main text the reader can be directed to exactly the section of supplemental text they need to look (e.g. you will say see supplemental notes section 2 etc).

Thank you for this comment. We had previously suggested to include especially more of the supplementary information into the introduction and then shift "supplementary" results into the main text. We agree that it is distracting as is and are more than happy to improve readability of the paper. In the revised version, we have shifted former supplementary notes, results and discussion into the main text.

3) Authors refer supplemental figures 8 and 9, which are non-existent (likely reflecting the fact the paper has been reformatted and reshuffled from previous submissions). Authors should carefully check that all cross referencing is correct. For example in supplemental notes when discussing cassette K4, it refers to figure 1C, but cassette K4 does not appear in 1C. In another part, in the main text authors refer to the MD simulations and guide the reader to sup figure 5, sup table and sup discussion. However there is a lot of information on the MD work in supplemental notes which is not cross-referenced here.

We very much apologize for these mistakes: Wrong referencing of Supplementary Figures 8 and 9 occurred in the list of Supplementary Information which was inserted for orientation at the very end of the manuscript and in the Supplementary

Notes part about the MD simulations. The wrongly annotated "Supplemental figures 8 and 9" refer to Supplementary Figure 5e-i and Supplementary Figure 6, respectively.

We have worked on the MD part and made the links to the Supplementary Information including the Supplementary Table 2 and Supplementary Note 2.

Cassette K4 was originally part of Fig. 1C but then shifted to a new Supplementary Figure 2 to reduce shown data as much as possible in the main figures. We eliminated the one reference to Figure 1c from the previous Supplementary Note which was incorporated in the main text now. Again, apologies for this mistake.

We have corrected all this and carefully double-checked annotations of all display items and references.

4) When saying "Neither TTG protein nor transcript levels responded to temperature shifts" authors should clarify that they mean TTG WITHOUT the lt-degron (i.e. 'normal' or 'WT' TTG), as this was confusing.

Thank you for pointing this out. We have adjusted this: *"Neither the endogenous TTG1 protein without the lt-degron nor TTG1 transcript levels responded to the temperature shifts when tested in a transgenic ttg1 TTG1-HA complementation line."*

5) Figure 1e - this figure is reported to show the 'tunable' nature of the degron. I understand the concept being conveyed - that, intermediate temperatures between permissive and restrictive temperatures lead to intermediate levels of accumulation. But to me it is not clear what was actually done in this figure. i.e. at what actual timepoint was the shift from 13 to 25 made (the arrow merely shows a gradation between the two), and at which point is this tunability demonstrated? Is this figure supposed to show that at 25 the band intensity is less than seen at 13, but still more than seen at 29 (as shown in figure 1d)? Perhaps the figure and experiment need a bit more explanation.

We thank the reviewer for suggesting a clarification here. We have included a more detailed description of the experiments in both the figure caption and the main text now.

We have shifted plants at $t = 0$ from 13 to 25°C (upper panel) or kept at 13°C (lower panel). Therefore, we are observing the gradual decrease of lt-protein over time until steady-state levels of translation versus degradation are reached and lead to presence of a certain "dose" of lt-protein. Fig. 1e demonstrates that we can adjust and keep protein abundance at a certain level that is lower than under permissive (cold) conditions and higher than under restrictive (warm) conditions.

The interpretation of "tunability" in this context is correct, i.e. intermediate temperatures lead to intermediate protein levels and eventually also activity. The latter is actually now also shown by new additional data using lt-GUS (Fig. 4h). In this second data set demonstrating tunability, we performed cold-warm-cold as well as the reciprocal shifts in 24 h

intervals, i.e. keeping the same temperature for 24 h for each period before continuing to shift further up or down. In this case, levels of active protein, as indicated by the measured GUS activity, can be efficiently tuned which highlights the possibilities in terms of tuning protein function by using the lt-degron system.

6) When referring to the GFP degradation dynamics, it is written "and both processes were tracked in time course experiments". What 'processes' are being referred to here?

This part was indeed unclear. For K2:GFP, we have designed two lt-degron cassettes with different N-terminal amino acids to test for degradation via the two known branches of the N-end rule pathway. We used either Arg or Phe as N-termini. These are supposed to be recognized by the two known N-recognins, i.e. Arg via PROTEOLYSIS6 (PRT6) and Phe via PRT1. We have improved this part in the revised version:

"Moreover, for K2:GFP, we have designed two lt-degron cassettes with different N-terminal amino acids to test for degradation via the two known branches of the N-end rule pathway. We used either the previously mentioned Phe or, as a new potentially destabilizing residue, Arg which results as neo-N-terminal after deubiquitination of the translated K2:GFP fusion protein. These residues are supposed to be recognized by the two known N-recognins, i.e. Arg via PROTEOLYSIS6 (PRT6) and Phe via PRT1. The response of these two constructs was compared under standard growth conditions (Figure S4c) and in time-course experiments, to demonstrate robustness (Figure S4d)."

7) Supplemental figure 4, I assume #F6 and #F41 etc refers to independent transgenic lines? This should be stated in the legend.

This is absolutely correct, we should have included this information which is now in the caption:

"#F6, #F41, #R42, and #R45 refers to independent transgenic lines with initiating Phe (F) or Arg (R) residues at the neo-N-terminal after deubiquitination, respectively."

8) In Figure 4D, what is the lower band seen in the K2:GUS anti-HA blot at 29°C that is not present at the lower temperature?

To determine the identity of the lower, a bit "misbehaving" band proved challenging. It runs at predicted size for the entire lt-GUS fusion protein in an SDS-PAGE, the upper band, however, responds to the temperature shifts. Presence as well as accumulation of the upper band are directly linked to GUS activity, see previous Figures 4d/e and f/g. We think that the lower band is actually a degradation product containing immunogenic parts of GUS full length. Importantly, treatment with proteasome inhibitor leads to accumulation only of the upper but not of the lower band (Supplementary Figure 4f). To clarify whether DHFR from the lt-degron fusion partner is in both or only one band, we analyzed parts of an SDS-PAGE gel by mass spectrometry (Supplementary Figure 7).

We found peptides corresponding to GUS and DHFR only in the upper, responsive band indicating the presence of the full length lt-GUS containing the N-degron. Some GUS peptides,

however, were also detected in the lower band but DHFR could not be found here. This suggests that the lower band might be due to degradation, breakage or could be devoid of the DHFR tag for other reasons.

9) Figure 4 e and f: I do not see the need to show both logarithmic and non-logarithmic versions of these data, especially when the differences are not clear in the logarithmic version (4e). I think that 4f alone is sufficient as this shows the significant reduction in GUS activity upon transition to the restrictive temperature.

Fig. 4e shows the non-responsiveness of the very strong 35S::GUS signal which has even a bit stronger activity at warm temperature, highlighting the effectiveness of the lt-degron. Therefore, we designed two panels. However, we agree that this figure can be streamlined and we have adjusted the figure (Figure 4e) according to the suggestions.

10) In Figure 7a, why do GFP levels decrease upon CHX treatment even at the permissive temperature (albeit not complete depletion as seen at the restrictive temperature)? Also in Figure 7c, to me these blots are not entirely convincing as the protein signal is so faint.... In contrast the whole fly data in figure 7d are very nice and convincing.

We think that the supposed incomplete accumulation in Fig. 7a and c is due to 1) residual proteolytic activity of the system and 2) also the nature of the experiment, i.e. tracking transiently expressed plasmid-born transgenes after transformation in cell culture. We are working in a steady-state system and the lt-protein is always translated but only conditionally degraded. Therefore, the detected levels depend on the time after transfection which was relatively short before the CHX treatment. This possibly explains the lower levels of lt-TEV even under cold conditions. The difference in protein levels between lane one and three (cold, mock-treated) and lane two (cold, CHX treatment) can therefore be possibly explained as follows. 1) Protein synthesis for 4h of the treatment that still occurs in sample for lane one and three but not for the sample of lane two due to CHX treatment. 2) We think that also under permissive conditions, the lt-degron fusion can be instable and be degraded dependent on the fusion partner. This is demonstrated by the new Figure 1f where we show that the lt-degron fusion is stabilized under warm conditions when introgressed into the *prt1* mutant background. In the *prt1* mutant, accumulation of fusion protein also occurs to a certain extent under cold conditions. For a detailed description see below. 3) Strong overexpression in cell culture is also likely to result in degradation by other proteolytic pathways.

We have included in our response another western blot with clearer results demonstrating that application of the system

is also useful in this system.

In Fig. 7a - c, we actually show data from two systems (two different embryonic cell cultures), which we initially intended to avoid by making the lt-degron available to whole organisms as shown by the fly work in Fig. 7d.

REVIEWERS' COMMENTS:

Reviewer #4 (Remarks to the Author):

In this paper, Dissmeyer and colleagues report the development of a generic tool for conditional protein destabilisation based on temperature, with lower ('permissive') temperatures facilitating protein stability and higher ('restrictive') temperatures promoting destabilisation, thus allowing 'tunable' protein accumulation. The system described is based on a previously reported construct that worked only in single celled organisms at higher temperatures that were not appropriate for use in the vast majority of multicellular plants and animals. Here, Dissmeyer et al have analysed a series of variants of this original construct and identified one that works at temperatures within the range that is appropriate for use in a wide range of multicellular organisms, which they term the low-temperature degron (It-degron). The authors have gone to great lengths to demonstrate the wide ranging applicability of this new method for 'on demand' protein accumulation or depletion in a large range of eukaryotic organisms and for functionally diverse proteins. They have shown its use in Arabidopsis, tobacco, yeast and drosophila, which supports their claims that this system could benefit the wider biological scientific community. Impressively, they not only demonstrated temperature dependent degradation/stability for a wide range of proteins (including GUS, GFP, TEV, URA3), but in Arabidopsis they linked conditional protein accumulation to direct developmental phenotypes; TTG bearing the It-N-degron was shown promote trichome development in a ttg mutant only at permissive temperatures, whilst It-degron CONSTANS promoted floral transition also only at the permissive temperature. I believe that Dissmeyer et al have developed - and successfully demonstrated the potential widespread utility of - a novel and important tool for condition-responsive protein accumulation that is of interest to the wider scientific community.

I have a number of relatively minor comments that need to be addressed:

- 1) In general, the text in the figures (e.g. labels of blots etc) is quite small, and should be increased in size to make them clearer
- 2) Particularly early on in the paper there is a lot of information that has been placed in supplemental, which was often quite confusing. Can I suggest that each separate subsection of e.g. supplemental notes (and later on, supplemental discussion) be numbered, so that in the main text the reader can be directed to exactly the section of supplemental text they need to look (e.g. you will say see supplemental notes section 2 etc).
- 3) Authors refer supplemental figures 8 and 9, which are non-existent (likely reflecting the fact the paper has been reformatted and reshuffled from previous submissions). Authors should carefully check that all cross referencing is correct. For example in supplemental notes when discussing cassette K4, it refers to figure 1C, but cassette K4 does not appear in 1C. In another part, in the main text authors refer to the MD simulations and guide the reader to sup figure 5, sup table and sup discussion. However there is a lot of information on the MD work in supplemental notes which is not cross-referenced here.
- 4) When saying "Neither TTG protein nor transcript levels responded to temperature shifts" authors should clarify that they mean TTG WITHOUT the It-degron (i.e. 'normal' or 'WT' TTG), as this was confusing.
- 5) Figure 1e - this figure is reported to show the 'tunable' nature of the degron. I understand the concept being conveyed - that, intermediate temperatures between permissive and restrictive temperatures lead to intermediate levels of accumulation. But to me it is not clear what was actually

done in this figure. i.e. at what actual timepoint was the shift from 13 to 25 made (the arrow merely shows a gradation between the two), and at which point is this tunability demonstrated? Is this figure supposed to show that at 25 the band intensity is less than seen at 13, but still more than seen at 29 (as shown in figure 1d)? Perhaps the figure and experiment need a bit more explanation.

6) When referring to the GFP degradation dynamics, it is written "and both processes were tracked in time course experiments". What 'processes' are being referred to here?

7) Supplemental figure 4, I assume #F6 and #F41 etc refers to independent transgenic lines? This should be stated in the legend.

8) In Figure 4D, what is the lower band seen in the K2:GUS anti-HA blot at 29°C that is not present at the lower temperature?

9) Figure 4 e and f: I do not see the need to show both logarithmic and non-logarithmic versions of these data, especially when the differences are not clear in the logarithmic version (4e). I think that 4f alone is sufficient as this shows the significant reduction in GUS activity upon transition to the restrictive temperature.

10) In Figure 7a, why do GFP levels decrease upon CHX treatment even at the permissive temperature (albeit not complete depletion as seen at the restrictive temperature)? Also in Figure 7c, to me these blots are not entirely convincing as the protein signal is so faint.... In contrast the whole fly data in figure 7d are very nice and convincing.